# TOIST: Task Oriented Instance Segmentation Transformer with Noun-Pronoun Distillation

**Pengfei Li[1]**    **Beiwen Tian[1]**    **Yongliang Shi[1]**    **Xiaoxue Chen[1]**
**Hao Zhao[2,3]**    **Guyue Zhou[1]**    **Ya-Qin Zhang[1]**
[1]AIR, Tsinghua University    [2] Peking University    [3]Intel Labs
li-pf22@mails.tsinghua.edu.cn
zhao-hao@pku.edu   hao.zhao@intel.com

## Abstract

Current referring expression comprehension algorithms can effectively detect or segment objects indicated by nouns, but how to understand verb reference is still under-explored. As such, we study the challenging problem of task oriented detection, which aims to find objects that best afford an action indicated by verbs like *sit comfortably on*. Towards a finer localization that better serves downstream applications like robot interaction, we extend the problem into task oriented instance segmentation. A unique requirement of this task is to select *preferred* candidates among possible alternatives. Thus we resort to the transformer architecture which naturally models pair-wise query relationships with attention, leading to the TOIST method. In order to leverage pre-trained noun referring expression comprehension models and the fact that we can access privileged noun ground truth during training, a novel noun-pronoun distillation framework is proposed. Noun prototypes are generated in an unsupervised manner and contextual pronoun features are trained to select prototypes. As such, the network remains noun-agnostic during inference. We evaluate TOIST on the large-scale task oriented dataset COCO-Tasks and achieve +10.9% higher $\mathrm{mAP^{box}}$ than the best-reported results. The proposed noun-pronoun distillation can boost $\mathrm{mAP^{box}}$ and $\mathrm{mAP^{mask}}$ by +2.8% and +3.8%. Codes and models are publicly available at https://github.com/AIR-DISCOVER/TOIST.

## 1   Introduction

As benchmarked by the RefCOCO, RefCOCO+ [27][61] and RefCOCO-g [45] datasets, noun referring expression comprehension models have seen tremendous progress, thanks to large-scale vision-language pre-training models like VL-BERT [57], VilBERT [43], OSCAR [33], UNITER [10], 12-in-1 [44] and MDETR [26]. As shown in the left top part of Fig. 1, these algorithms take noun prompts like *hatchback car* as inputs and generate a bounding box or an instance mask of that car. However, in real-world applications like intelligent service robots, system inputs usually come in the form of *affordance* (i.e., the capability to support an action or say a verb phrase). Whether modern vision-language model designs can effectively understand verb reference remains under-explored.

To this end, we focus on the challenging problem of task oriented detection, as introduced by the COCO-Tasks benchmark [55]. As shown in the right top part of Fig. 1, a task oriented detector outputs three boxes of forks as they can be used to *smear butter*. We also extend the problem to an upgraded instance segmentation version using existing COCO masks [40], as the masks can provide finer localization. When RGB-D pairs are available, instance masks can be used to obtain object point clouds. When image sequences are available, instance masks can be used to reconstruct objects using visual hull [31][11][67]. As such, the newly proposed task oriented instance segmentation formulation (Fig. 1 bottom) is useful for down-stream robot interaction applications.

36th Conference on Neural Information Processing Systems (NeurIPS 2022).

While noun referring expression comprehension datasets aim to minimize ambiguity [45], an interesting and challenging feature of task oriented detection/segmentation is the intrinsic ambiguity. For example, in the right top panel of Fig. 1, the pizza peel can also be used to *smear butter*. If we have neither forks nor pizza peels at hand, it is still possible to use the plate to *smear butter*. Another example is shown in Fig. 1 bottom. When we consider an object to *step on*, the chair is a better choice because the sofa is soft and the table is heavy to move. When the need switches to *sit comfortably on*, sofas are obviously the best candidates. In one word, objects that afford a verb are ambiguous and the algorithm needs to model *preference*.

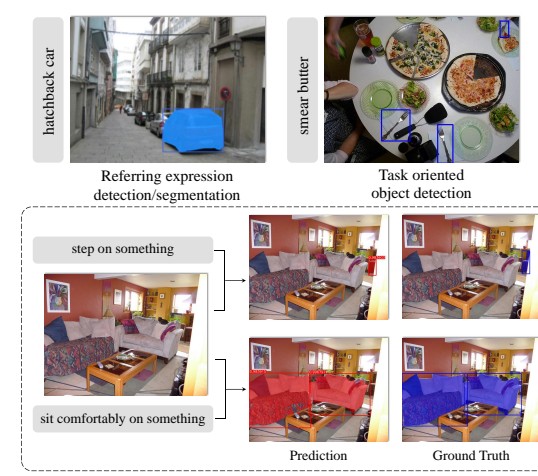

Figure 1: Left top: Noun referring expression comprehension. Right top: Task oriented detection. Bottom: Task oriented instance segmentation.

To this end, current models [55] use a two-stage pipeline, in which objects are firstly detected then relatively ranked. Inspired by the success of DETR-like methods [6][65][41] and the advantage of the attention mechanism in revealing the relationship between visual elements [34][64], we resort to the transformer architecture as it imposes self-attention on object queries thus naturally models the pair-wise relative preference between object candidates. Our one-stage method is named as **T**ask **O**riented **I**nstance **S**egmentation **T**ransformer and abbreviated as TOIST. Transformers are considered to be data hungry [5][14], but obtaining large-scale visually grounded verb reference data with relative preference (e.g., COCO-Tasks [55]) is difficult. This inspires us to explore the possibility of reusing knowledge in noun referring expression comprehension models. We propose to use pronouns like *something* as a proxy and distill knowledge from noun embedding prototypes generated by clustering.

Specifically, we first train a TOIST model with verb-noun input (e.g., *step on chair* for the bottom panel of Fig. 1 bottom), using the privileged noun ground truth. But during inference, we cannot access the noun *chair*, thus we train the second TOIST model with verb-pronoun input (e.g., *step on something*) and distill knowledge from the first TOIST model. As such, the second TOIST model remains noun-agnostic during inference and achieves better performance than directly training a model with verb-pronoun input. This framework is named as *noun-pronoun distillation*. Although leveraging knowledge from models with privileged information has been used in robotics research like autonomous driving [7] and quadrupedal locomotion [32], the proposed paradigm of distilling privileged noun information into pronoun features is novel, to the best of our knowledge.

To summarize, this paper has the following four contributions:

- We upgrade the task oriented detection task into task oriented instance segmentation and provide the first solution to it. Although this is a natural extension, this new formulation is of practical value to robotics applications.

- Unlike existing two-stage models that firstly detect objects then rank them, we propose the first transformer-based method TOIST, for task oriented detection/segmentation. It has only one stage and naturally models relative preference with self-attention on object queries.

- In order to leverage the privileged information in noun referring expression understanding models, we propose a novel noun-pronoun distillation framework. It improves TOIST by +2.8% and +3.8% for $\mathrm{mAP^{box}}$ and $\mathrm{mAP^{mask}}$, respectively.

- We achieve new state-of-the-art (SOTA) results on the COCO-Tasks dataset, out-performing the best reported results by +10.9% $\mathrm{mAP^{box}}$. Codes and models are publicly available.

## 2 Related Works

**Vision and Language.** Connecting vision and language is a long-existing topic for visual scene understanding. Barnard et al. [4] propose a system that translates image regions into nouns. Babytalk

[30] is an early method that turns images into sentences, based upon conditional random fields. Visual question answering [3] aims to answer questions about an image, with potential applications in helping visually impaired people [19]. The CLEVR dataset [25] focuses on the reasoning ability of question answering models, thanks to a full control over the synthetic data. The Flickr30k benchmark [51] addresses the phrase grounding task that links image regions and descriptions. Vision-language navigation [2][16] aims to learn navigation policies that fulfill language commands. DALL-E [52] shows impressive text-to-image generation capability. Video dense captioning [28][15] generates language descriptions for detected salient regions. Visual madlibs [60] focuses on fill-the-blank question answering. Visual commonsense reasoning [62] proposes the more challenging task of justifying an answer. Referring expression detection [61][45] and segmentation [24] localize objects specified by nouns. Although this literature is very large with many problem formulations proposed, the task of detecting objects that afford *verbs* (e.g., COCO-Tasks[55]) is still under-explored.

**Action and Affordance.** Verbs, as the link between subjects and objects, have been extensively studied in the both vision [58][48] and language [23][46] communities before, while we focus on the vision side. It is difficult to define standalone *verb recognition* tasks, so existing problem formulations depend on the focus on subjects or objects. A simple taxonomy can be considered as such: recognizing *subjects and verbs* is named as action recognition [56][17]; recognizing *verbs and objects* is named as affordance recognition [66][59][13][9]; recognizing *triplets* is named as human-object-interaction recognition [35][63]. Task oriented object detection/segmentation, as an affordance understanding task, is very challenging and this study explores the noun-pronoun distillation framework to borrow rich knowledge from more visually grounded noun targets.

**Knowledge Distillation.** This technique is proposed in the deep learning literature by Hinton et al. [22] to distill knowledge from large models to small models. The insight is that soft logit targets generated by large models contain richer information that better serves as a supervision signal than hard one-hot labels. Knowledge distillation has been extended to show effectiveness in other domains like continual learning [37] and object detection [8]. The survey of Guo et al. [18] provides a comprehensive summary of knowledge distillation variants and applications. Most related to our method is the privileged knowledge distillation methods in robotics research [7][32], in which the teacher model has access to privileged information that the student cannot access during inference. Our noun-pronoun distillation method is tailored for the task oriented detection/segmentation problem which borrows rich knowledge from noun referring expression compression teacher models while still allowing the student model to be noun-agnostic.

## 3   Formulation

The problem is to detect and segment objects that are preferred to afford a specific task indicated by verb phrases, from an input image. Yet clearly defining **affordance** and **preference** is actually challenging so we follow the existing annotation protocol of the COCO-Tasks dataset [55]:

**Affordance.** Firstly, the target objects *afford* a specific task. In an input image, it is possible that no objects or multiple objects afford the task. And in the latter case, the objects may belong to multiple classes. For example, in the right top panel of Fig.1, nothing affords the task *sit comfortably on*. Instead, in the bottom panel, there are at least two sofas, a table and a chair that afford the task.

**Preference.** Secondly, we need to find the *best* ones from the objects which afford the task. In other words, the *preference* among multiple objects needs to be understood. In Fig.1 bottom, the two sofas are obviously more suitable for the task *sit comfortably on* than other objects enumerated above. Thus the ground truth objects for this task are the two sofas (covered in blue).

Now we formally define the task. The input is an RGB image $\mathbf{X}_v \in \mathbb{R}^{3 \times H_0 \times W_0}$ ($v$ represents *visual*) and a piece of text $\mathbf{X}_l$ ($l$ represents *language*). $\mathbf{X}_l$ describes a specific task like *sit comfortably on*. The targets are bounding boxes $\mathbf{B}_{\mathrm{gt}} = [b_1, \ldots, b_{n_{\mathrm{gt}}}] \in [0,1]^{n_{\mathrm{gt}} \times 4}$ and instance segmentation masks $\mathbf{M}_{\mathrm{gt}} = [m_1, \ldots, m_{n_{\mathrm{gt}}}] \in \mathbb{R}^{n_{\mathrm{gt}} \times H_0 \times W_0}$ of target objects $\mathbf{O}_{\mathrm{gt}} = \langle \mathbf{B}_{\mathrm{gt}}, \mathbf{M}_{\mathrm{gt}} \rangle$, where $n_{\mathrm{gt}} \geq 0$ is the count of targets. The four components of $b_i \in [0,1]^4$ are normalized center coordinates, height and width of the $i$-th box. An algorithm $f$ that addresses this problem works as:

$$f(\langle \mathbf{X}_v, \mathbf{X}_l \rangle) = \langle \mathbf{B}_{\mathrm{pred}}, \mathbf{M}_{\mathrm{pred}}, \mathbf{S}_{\mathrm{pred}} \rangle, \tag{1}$$

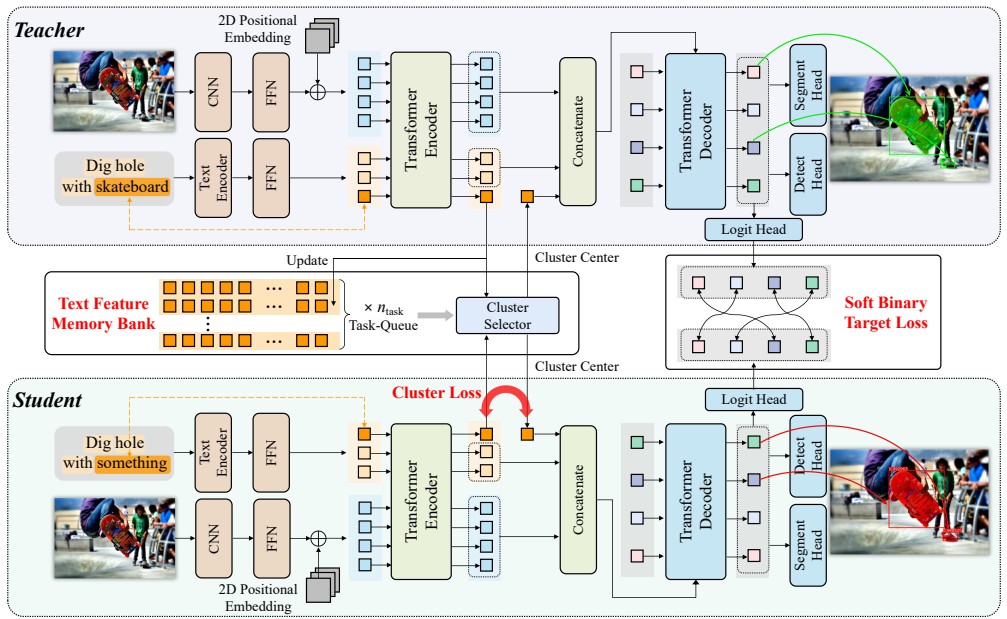

Figure 2: TOIST network architecture and the noun-pronoun distillation framework. The cluster loss and soft binary target loss distill privileged noun knowledge and preference knowledge respectively.

where $\mathbf{B}_{\text{pred}}$ and $\mathbf{M}_{\text{pred}}$ are predicted boxes and masks, respectively. $\mathbf{S}_{\text{pred}} = [\hat{s}_1, \ldots, \hat{s}_{n_{\text{pred}}}] \in [0, 1]^{n_{\text{pred}}}$ is the probability scores of the predicted objects being selected, which reflects preference. We denote the predicted objects as $\mathbf{O}_{\text{pred}} = \langle \mathbf{B}_{\text{pred}}, \mathbf{M}_{\text{pred}}, \mathbf{S}_{\text{pred}} \rangle$.

## 4    Method

We propose an end-to-end **T**ask **O**riented **I**nstance **S**egmentation **T**ransformer, abbreviated as TOIST (Section 4.1). Leveraging pre-trained noun referring expression comprehension models, we further adopt a teacher-student framework for noun-pronoun distillation (Section 4.2).

### 4.1    Task Oriented Instance Segmentation Transformer

The SOTA method [55] uses a two-stage pipeline to solve the problem. Taking a single image $\mathbf{X}_v$ as input, it first detects the bounding boxes $\mathbf{B}_{\text{pred}}$ of all objects with a Faster-RCNN [53]. Then it ranks $\mathbf{B}_{\text{pred}}$ with a GNN [36], predicting the probabilities $\mathbf{S}_{\text{pred}}$ of the objects being selected for a task. Our method differs from it in three ways: (1) We address the task with a one-stage architecture, allowing joint representation learning for detection and preference modeling. (2) We specify tasks using the text $\mathbf{X}_l$. (3) We predict instance masks $\mathbf{M}_{\text{pred}}$ along with bounding boxes.

We choose to build our method upon the transformer architecture [6], because the self-attention operators in the decoder can naturally model pair-wise relative preference between object candidates. As shown in Fig.2 bottom, TOIST contains three main components: a multi-modal encoder (color brown) to extract tokenized features, a transformer encoder (color green) to aggregate features of two modalities and a transformer decoder (color blue) to predict the most suitable objects with attention.

**Two Input Forms.** To find an object that affords the task of *dig hole*, the first step is to construct a task description input $\mathbf{X}_l$. To achieve this goal, we can extend the task name with the ground truth object category to the **verb-noun form** like *dig hole with skateboard* or with a pronoun to the **verb-pronoun form** like *dig hole with something*. While the former violates the noun-agnostic constraint during inference, it can be leveraged to improve the latter within the proposed noun-pronoun distillation framework, which will be detailed later. For a plain TOIST, the verb-pronoun form is selected as task description $\mathbf{X}_l$, which is fed into the multi-modal encoder along with visual input $\mathbf{X}_v$.

**Multi-Modal Encoder.** For $\mathbf{X}_v$, a pre-trained CNN-based backbone and a one-layer feed forward network (FFN) are leveraged to extract a low-resolution feature map $\mathbf{F}_v \in \mathbb{R}^{d \times H \times W}$. Flattening the spatial dimensions of $\mathbf{F}_v$ into one dimension, we obtain a sequence of tokenized feature vectors $\mathbf{V} = [v_1, \ldots, v_{n_v}] \in \mathbb{R}^{n_v \times d}$ (light blue squares ■ on the left of the transformer encoder in Fig.2), where $n_v = H \times W$. To preserve the spatial information, 2D positional embeddings are added to $\mathbf{V}$. For $\mathbf{X}_l$, we use a pre-trained text encoder and another FFN to produce corresponding feature vectors $\mathbf{L} = [l_1, \ldots, l_{n_l}] \in \mathbb{R}^{n_l \times d}$ (light orange squares ■), where $n_l$ is the total count of language tokens. Among these $n_l$ features, we denotes the one corresponding to the pronoun (or noun) token as $l_{\text{pron}}$ (or $l_{\text{noun}}$) (dark orange squares ■). We concatenate these vectors and obtain the final feature sequence $[\mathbf{V}, \mathbf{L}] = [v_1, \ldots, v_{n_v}, l_1, \ldots, l_{n_l}] \in \mathbb{R}^{(n_v + n_l) \times d}$.

**Transformer Encoder.** The transformer encoder consists of $n_{\text{tr}}$ sequential blocks of multi-head self-attention layers. Given the sequence of features $[\mathbf{V}, \mathbf{L}]$, it outputs processed feature sequence $[\mathbf{V}^{\text{tr}}, \mathbf{L}^{\text{tr}}] = [v_1^{\text{tr}}, \ldots, v_{n_v}^{\text{tr}}, l_1^{\text{tr}}, \ldots, l_{n_l}^{\text{tr}}] \in \mathbb{R}^{(n_v + n_l) \times d}$. The pronoun (or noun) feature $l_{\text{pron}}$ (or $l_{\text{noun}}$) is encoded into $l_{\text{pron}}^{\text{tr}}$ (or $l_{\text{noun}}^{\text{tr}}$), which will be used for noun-pronoun distillation (Section 4.2). Here for the plain TOIST, we directly use the features $[\mathbf{V}^{\text{tr}}, \mathbf{L}^{\text{tr}}]$ for later processing.

**Transformer Decoder.** The transformer decoder consists of $n_{\text{tr}}$ blocks of self-attention and cross-attention layers. It takes as input a set of learnable parameters serving as an object query sequence $\mathbf{Q} = [q_1, \ldots, q_{n_{\text{pred}}}] \in \mathbb{R}^{n_{\text{pred}} \times d}$. $[\mathbf{V}^{\text{tr}}, \mathbf{L}^{\text{tr}}]$ are used as keys and values for cross-attention layers. The outputs of the transformer decoder are feature vectors $\mathbf{Q}^{\text{tr}} = [q_1^{\text{tr}}, \ldots, q_{n_{\text{pred}}}^{\text{tr}}] \in \mathbb{R}^{n_{\text{pred}} \times d}$, which are projected to final results by three prediction heads. Specifically, the detect head predicts bounding boxes $\mathbf{B}_{\text{pred}}$ and the segment head outputs binary segmentation masks $\mathbf{M}_{\text{pred}}$. The logit head outputs logits $\mathbf{G}_{\text{pred}} = [\hat{\mathbf{g}}_1, \ldots, \hat{\mathbf{g}}_{n_{\text{pred}}}] \in \mathbb{R}^{n_{\text{pred}} \times n_{\max}}$, where $\hat{\mathbf{g}}_i = [\hat{g}_1^i, \ldots, \hat{g}_{n_{\max}}^i]$. $[\hat{g}_1^i, \ldots, \hat{g}_{n_l}^i]$ corresponds to text tokens $\mathbf{L} = [l_1, \ldots, l_{n_l}] \in \mathbb{R}^{n_l \times d}$. $[\hat{g}_{n_l+1}^i, \ldots, \hat{g}_{n_{\max}-1}^i]$ is used to pad $\hat{\mathbf{g}}_i$ to length $n_{\max}$ (by default $n_{\max}$=256) and the last one $\hat{g}_{n_{\max}}^i$ stands for the logit of "no-object". With the output logits, we define the preference score $\hat{s}_i \in \mathbf{S}_{\text{pred}}$ of each predicted object as:

$$\hat{s}_i = 1 - \frac{\exp\left(\hat{g}_{n_{\max}}^i\right)}{\sum_{j=1}^{n_{\max}} \exp\left(\hat{g}_j^i\right)}. \tag{2}$$

During training, as in DETR [6], a bipartite matching is computed between $n_{\text{pred}}$ predicted objects $\mathbf{O}_{\text{pred}}$ and ground truth objects $\mathbf{O}_{\text{gt}}$ with the Hungarian algorithm [29]. The matched object predictions are supervised with L1 loss and Generalized Intersection over Union (GIoU) loss [54] for localization while Dice/F-1 loss [47] and Focal cross-entropy loss [39] for segmentation. We also adopt the soft-token prediction loss and the contrastive alignment loss used in MDETR [26]. But different from them, we do not use a single noun or pronoun as the ground truth token span for a matched object prediction. Instead, we use the whole verb-pronoun description as token span such that the network can understand the verbs rather than noun/pronoun only. The total loss for TOIST is:

$$\mathcal{L}_{\text{TOIST}} = \lambda_1 \mathcal{L}_{\text{l1}} + \lambda_2 \mathcal{L}_{\text{giou}} + \lambda_3 \mathcal{L}_{\text{dice}} + \lambda_4 \mathcal{L}_{\text{cross}} + \lambda_5 \mathcal{L}_{\text{token}} + \lambda_6 \mathcal{L}_{\text{align}}, \tag{3}$$

where $\lambda_1 \sim \lambda_6$ are the weights of losses.

## 4.2 Noun-Pronoun Distillation

**Motivation.** As mentioned above, there are two possible input forms. Due to the privileged information of target names (nouns), TOIST using verb-noun input out-performs its counterpart by +11.8% and +12.0% on $\text{mAP}^{\text{box}}$ and $\text{mAP}^{\text{mask}}$, as demonstrated in Table. 1. We use two preliminary experiments as a motivation: directly replacing the pronoun feature $l_{\text{pron}}$ or $l_{\text{pron}}^{\text{tr}}$ in the verb-pronoun model with the corresponding noun feature $l_{\text{noun}}$ or $l_{\text{noun}}^{\text{tr}}$ in the verb-noun

Table 1: TOIST quantitative results under several different settings related to text.

| Text related settings | $\text{mAP}^{\text{box}}$ | $\text{mAP}^{\text{mask}}$ |
|---|---|---|
| verb-pronoun input | 41.3 | 35.2 |
| verb-noun input | 53.1 (+11.8) | 47.2 (+12.0) |
| replace $l_{\text{pron}}$ with $l_{\text{noun}}$ | 43.7 (+2.4) | 37.3 (+2.1) |
| replace $l_{\text{pron}}^{\text{tr}}$ with $l_{\text{noun}}^{\text{tr}}$ | 41.9 (+0.6) | 35.6 (+0.4) |
| noun-pronoun distillation | **44.1 (+2.8)** | **39.0 (+3.8)** |

model can boost performance (see Table 1). However, during inference, the noun of the ground truth object is unavailable. Thus we believe a properly designed noun-pronoun distillation framework can leverage rich knowledge from the verb-noun model without violating the noun-agnostic constraint.

**Distillation Framework Overview.** Two TOIST models are trained simultaneously. The teacher (Fig.2 top) and the student (Fig.2 bottom) take as input verb-noun and verb-pronoun descriptions, respectively. A clustering distillation method with a memory bank and a tailored cluster loss is used to distill privileged object-centric knowledge from noun to pronoun (Fig.2 middle left). Besides, we also use a soft binary target loss imposed on $\mathbf{G}_{\text{pred}}$ to distill preference knowledge (Fig.2 middle right), in which $\mathbf{G}_{\text{pred}}$ are logits used to calculate preference scores $\mathbf{S}_{\text{pred}}$.

**Clustering Distillation.** Since one task can be afforded by objects of many different categories, we build a text feature memory bank to store noun features, with which a prototype can be selected and used to replace pronoun feature and distill knowledge. We term this process as clustering distillation. Specifically, we use $l_{\text{pron}}^{\text{tr}}$ and $l_{\text{noun}}^{\text{tr}}$ instead of $l_{\text{pron}}$ and $l_{\text{noun}}$ for this process. The reason is that the former ones are conditioned on the image input and verb tokens of the task by self-attention layers, and thus it is meaningful to select a cluster center that suits the image and the task input.

**Memory and Selector.** The size of the memory bank is $n_{\text{task}} \times n_{\text{mem}} \times d$. It consists of $n_{\text{task}}$ queues of length $n_{\text{mem}}$ for $n_{\text{task}}$ tasks. During training, for each sample from task $j$, we update the $j$-th queue $\mathbf{L}_{\text{mem}}^j = [l_1^j, \ldots, l_{n_{\text{mem}}}^j]$ by adding the noun feature $l_{\text{noun}}^{\text{tr}}$ generated by the teacher model and removing the existing one closest to $l_{\text{noun}}^{\text{tr}}$. The updated queue is clustered with the K-means clustering method, leading to K cluster centers $\mathbf{L}_c^j = \{l_{c_1}^j, \ldots, l_{c_K}^j\}$. Then the student model uses a cluster selector, which is implemented as the nearest neighbor classifier, to select a prototype $l_{c_s}^j \in \mathbf{L}_c^j$ according to the pronoun feature $l_{\text{pron}}^{\text{tr}}$ and replace $l_{\text{pron}}^{\text{tr}}$ with $l_{c_s}^j$. Concatenating other tokens and the selected prototype together, the output of student transformer encoder $[v_{s_1}^{\text{tr}}, \ldots, v_{s_n}^{\text{tr}}, l_{s_1}^{\text{tr}}, \ldots, l_{\text{pron}}^{\text{tr}}, \ldots, l_{s_{n_l}}^{\text{tr}}]$ is modified into $[v_{s_1}^{\text{tr}}, \ldots, v_{s_n}^{\text{tr}}, l_{s_1}^{\text{tr}}, \ldots, l_{c_s}^j, \ldots, l_{s_{n_l}}^{\text{tr}}]$ and fed into the transformer decoder. To distill knowledge to the student transformer encoder, we define cluster loss as:

$$\mathcal{L}_{\text{cluster}} = \|l_{\text{pron}}^{\text{tr}} - l_{c_s}^j\|_2, \tag{4}$$

with which the privileged object-centric knowledge is distilled from clustered noun features to pronoun feature and further to the student TOIST encoder.

**Preference Distillation.** We use a soft binary target loss to distill preference knowledge from teacher to student. For an object query, we first define binary query probabilities of being positive-query or negative-query, which denotes whether or not the query is matched to a ground truth object target, as $\mathbf{p} = [p^{\text{pos}}, p^{\text{neg}}] \in \mathbb{R}^{1 \times 2}$. The probabilities can be calculated by the softmax function:

$$p^{\text{pos}} = \frac{\sum_{j=1}^{n_{\max}-1} \exp(\hat{g}_j)}{\sum_{j=1}^{n_{\max}} \exp(\hat{g}_j)}, p^{\text{neg}} = \frac{\exp(\hat{g}_{n_{\max}})}{\sum_{j=1}^{n_{\max}} \exp(\hat{g}_j)}, \tag{5}$$

where $\hat{g}_j$ and $\hat{g}_{n_{\max}}$ represent the logits corresponding to $i$-th text token and "no-object" token, respectively. For all the object queries in teacher and student, the probability sequences are denoted by $\mathbf{P}_t = [\mathbf{p}_{t_1}, \ldots, \mathbf{p}_{t_{n_{\text{pred}}}}]$ and $\mathbf{P}_s = [\mathbf{p}_{s_1}, \ldots, \mathbf{p}_{s_{n_{\text{pred}}}}]$. Then we use the Hungarian algorithm to find a bipartite matching between the two sequences of object queries with $\mathbf{P}_t$ and $\mathbf{P}_s$. Formally, we search for a permutation of $n_{\text{pred}}$ elements $\sigma \in \mathfrak{S}_{n_{\text{pred}}}$ which minimizes the matching loss:

$$\hat{\sigma} = \underset{\sigma \in \mathfrak{S}_{n_{\text{pred}}}}{\arg\min} \sum_i^{n_{\text{pred}}} \mathcal{L}_{\text{match}}(y_{t_i}, y_{s_{\sigma(i)}}), \tag{6}$$

where $y_{t_i} = (\hat{b}_{t_i}, \mathbf{p}_{t_i})$ and $\hat{b}_{t_i}$ is the predicted bounding box. $\mathcal{L}_{\text{match}}$ is a linear combination of box prediction losses (L1 & GIoU) and KL-Divergence. The KL-Divergence $\mathcal{L}_{\text{KL}}$ can be written as:

$$\mathcal{L}_{\text{KL}}(\mathbf{p}_{t_i}, \mathbf{p}_{s_{\sigma(i)}}) = \mathbf{KL}\left(\mathbf{p}_{t_i} \| \mathbf{p}_{s_{\sigma(i)}}\right) = p_{t_i}^{\text{pos}} \log\left(\frac{p_{t_i}^{\text{pos}}}{p_{s_{\sigma(i)}}^{\text{pos}}}\right) + p_{t_i}^{\text{neg}} \log\left(\frac{p_{t_i}^{\text{neg}}}{p_{s_{\sigma(i)}}^{\text{neg}}}\right). \tag{7}$$

With the optimal permutation $\hat{\sigma}$, we define the soft binary target loss as:

$$\mathcal{L}_{\text{binary}} = \sum_i^{n_{\text{pred}}} \mathcal{L}_{\text{KL}}(\mathbf{p}_{t_i}, \mathbf{p}_{s_{\hat{\sigma}(i)}}). \tag{8}$$

It makes the binary query probabilities of the student model similar to the matched ones of the teacher model. And because the preference score $\hat{s}$ (Eq.2) is defined in the same way as the probability $p^{\text{pos}}$ (Eq.5), the preference knowledge is distilled from teacher to student as the loss decreases.

Table 2: Comparison of the proposed method to SOTA baselines on the extended COCO-Tasks dataset. The methods with tags † and ‡ are used to calculate mAP margins. TOIST with noun-pronoun distillation achieves best results: +10.9% and +6.6% mAP on detection and segmentation.

| Object Detection | | Instance Segmentation | | |
|---|---|---|---|---|
| Method | $\text{mAP}^{\text{box}}$ | Method | $\text{mAP}^{\text{box}}$ | $\text{mAP}^{\text{mask}}$ |
| Faster-RCNN | 20.6 | Mask-RCNN | 23.4 | 20.0 |
| Faster-RCNN + pick best | 14.1 | Mask-RCNN + pick best | 18.8 | 16.8 |
| Faster-RCNN + ranker | 9.1 | Mask-RCNN + ranker | 10.6 | 9.3 |
| Faster-RCNN + classifier | 28.8 | Mask-RCNN + classifier | 33.7 | 29.2 |
| Faster-RCNN + GGNN | 32.6 | Mask-RCNN + GGNN‡ | 37.4 | 32.4 |
| Yolo + classifier | 29.1 | **TOIST** | **41.3** (+8.1) | **35.2** (+2.8) |
| Yolo + GGNN† | 33.2 | **TOIST w/ distillation** | **44.1** (+10.9) | **39.0** (+6.6) |

**Summary.** The final training loss function for TOIST with noun-pronoun distillation is:

$$\mathcal{L}_{\text{TOIST}-\text{NP}} = \mathcal{L}_{\text{TOIST}}^t + \mathcal{L}_{\text{TOIST}}^s + \lambda_7 \mathcal{L}_{\text{cluster}}^s + \lambda_8 \mathcal{L}_{\text{binary}}^s, \tag{9}$$

where $\lambda_7$ and $\lambda_8$ are the weights of losses. $\mathcal{L}_{\text{TOIST}}^t$ and $\mathcal{L}_{\text{TOIST}}^s$ are separate TOIST loss terms defined by Eq.3 for teacher and student, respectively. Note that the cluster loss $\mathcal{L}_{\text{cluster}}^s$ and the soft binary target loss $\mathcal{L}_{\text{binary}}^s$ are only used for supervising the student model.

As a reminder, during inference, we only use the student TOIST model and the fixed memory bank to find the most suitable objects, without violating the noun-agnostic constraint.

## 5  Experiments

**Dataset.** We conduct experiments on the COCO-Tasks dataset [55] which re-annotates the COCO dataset [40] with preference-aware affordance labels. This is the only dataset that involves instance-level preference in affordance. Though there are other datasets for affordance detection, they neither distinguish between instances nor involve preference, such as ADE-Affordance [12] and IIT-AFF [49]. The COCO-Tasks dataset contains 14 tasks. For each task, there are 3600 train images and 900 test images. In each image, the boxes of preferred objects (one or more) are taken as ground truth labels for detection. Using existing COCO masks, we extend the dataset to an instance segmentation version. In Appendix B and D, we present more dataset details and show its diversity.  **Metric.** We use the AP@0.5 metric for both detection and segmentation, where predicted preference scores $\mathbf{S}_{\text{pred}}$ are used to rank objects. Averaging the AP@0.5 values of all tasks leads to mAP@0.5. The implementation details for TOIST and distillation can be found in Appendix A.

### 5.1  Comparisons with State-of-the-art Methods

Table 2 shows that TOIST with noun-pronoun distillation achieves state-of-the-art results compared to existing methods on COCO-Tasks. For object detection, we use the results reported by [55] as baselines. For instance segmentation, following the same experiment settings of [55], we build new baselines using Mask-RCNN [20]. The methods in the first row treat the problem as a standard detection or segmentation task. All other baselines use two-stage pipelines, in which objects are firstly detected or segmented then ranked. The proposed one-stage method achieves 41.3% $\text{mAP}^{\text{box}}$ and 35.2% $\text{mAP}^{\text{mask}}$, which are +8.1% and +2.8% better than the previous best results (*Yolo+GGNN* and *Mask-RCNN+GGNN*). Noun-pronoun distillation further boosts the performance of TOIST to 44.1% (+10.9%) $\text{mAP}^{\text{box}}$ and 39.0% (+6.6%) $\text{mAP}^{\text{mask}}$. Our method also out-performs another

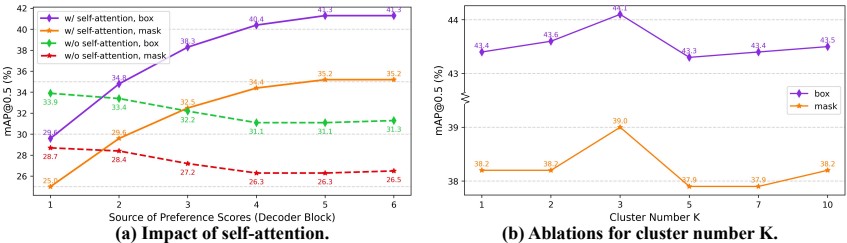

(a) Impact of self-attention.  (b) Ablations for cluster number K.

Figure 3: Experiments to study the impact of (a) self-attention and (b) cluster number.

baseline with the same backbone, as shown in Table 9 of Appendix C. These results demonstrate the effectiveness of the proposed method for the new problem of task oriented instance segmentation. Per-task quantitative results and precision-recall curves are provided in Appendix C and D.

## 5.2 Preference Modeling with Self-Attention

**Protocol.** Our design principle is that the self-attention layers in transformers can naturally model preference. But is this really the case? Fig.3 (a) shows its effect. Two plain TOIST models are trained separately, with the only difference being that one model does not contain self-attention operators in the decoder. Note that the removal of self-attention does not impact the number of parameters. The mAP metric is impacted by two kinds of errors: inaccurate box/mask localization or improper preference. To analyze the preference scores alone, for all object queries, we use the boxes and masks predicted by the last block of TOIST decoder, which are arguably the most accurate. We use the corresponding preference scores predicted by each block to calculate mAP values.

**Interpretation.** For the TOIST with self-attention, the performance is gradually boosted as the source of preference scores becomes deeper: from 29.6% $\text{mAP}^{\text{box}}$ and 25.0% $\text{mAP}^{\text{mask}}$ to 41.3% and 35.2%. For the one without self-attention, the preference scores from the first block lead to the best performance: 33.9% $\text{mAP}^{\text{box}}$ and 28.7% $\text{mAP}^{\text{mask}}$, which is -7.5% and -6.5% lower than another TOIST. The results demonstrate that the self-attention in TOIST decoder models pair-wise relative preference between object candidates. As the decoder deepens, the preference relationship between object candidates is gradually extracted by self-attention.

## 5.3 Effect of Clustering Distillation

In Table 3, we show the effects of using cluster loss and replacing pronoun features with cluster centers (noun prototypes). In (c) and (e), leveraging the two components alone brings an increase of +0.7% $\text{mAP}^{\text{box}}$, +1.9% $\text{mAP}^{\text{mask}}$ and +0.7% $\text{mAP}^{\text{box}}$, +1.8% $\text{mAP}^{\text{mask}}$ over baseline (a) respectively. In (g), the complete clustering distillation leads to a performance improvement of +1.0% $\text{mAP}^{\text{box}}$ and +2.3% $\text{mAP}^{\text{mask}}$. These results show that the clustering distillation method can improve student TOIST and enhance verb referring expression understanding.

Table 3: Ablations for distillation settings. CCR, CL and SBTL are short for cluster center replacement, cluster loss and soft binary target loss, respectively.

| Index | CCR | CL | SBTL | $\text{mAP}^{\text{box}}$ | $\text{mAP}^{\text{mask}}$ |
|-------|-----|-----|------|---------|----------|
| (a) | × | × | × | 41.3 | 35.2 |
| (b) | × | × | ✓ | 43.4 (+2.1) | 38.0 (+2.8) |
| (c) | × | ✓ | × | 42.0 (+0.7) | 37.1 (+1.9) |
| (d) | × | ✓ | ✓ | 43.8 (+2.5) | 38.6 (+3.4) |
| (e) | ✓ | × | × | 42.0 (+0.7) | 37.0 (+1.8) |
| (f) | ✓ | × | ✓ | 42.3 (+1.0) | 37.3 (+2.1) |
| (g) | ✓ | ✓ | × | 42.3 (+1.0) | 37.5 (+2.3) |
| (h) | ✓ | ✓ | ✓ | **44.1 (+2.8)** | **39.0 (+3.8)** |

In Fig.4, we visualize the predicted results (filtered by a preference threshold of 0.9) and the attention maps of pronoun tokens. In the first row, when there is no clustering distillation, TOIST wrongly prefers the flower to the cup, which is also confirmed by the attention map. But the TOIST with clustering distillation correctly selects the cup, and the attention on the flower is weakened. This shows that clustering distillation enables the student TOIST to reduce the ambiguity of verb-pronoun referring expression. In the second row, the bounding box of the knife is correctly detected by both two models. However, in the absence of the distillation, extra instance masks are predicted on the spoon and fork within the box. Instead, with the distillation, the masks predicted by TOIST are concentrated on the knife and the attention is more focused on it. This demonstrates that in the case of clustering distillation, TOIST can better ground the task into pixels within an object box.

Meanwhile, the fact that predicted masks may be inaccurate even if the box is correct makes it challenging for a robot to accurately grasp the preferred object when performing a specific task. This proves the importance of extending task oriented object detection to instance segmentation.

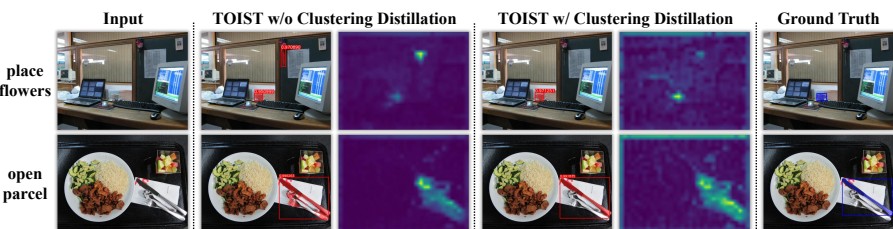

Figure 4: Visualization of the predicted results and attention maps of pronoun tokens.

## 5.4 Effect of Preference Distillation

In Table 3 (b), preference distillation with soft binary target loss achieves +2.1% $mAP^{box}$ and +2.8% $mAP^{mask}$ higher results than baseline (a). This loss acts on the preference probabilities of each object candidate in student TOIST. And the probabilities are used as scores to sort the object candidates for the calculation of mAP values. Therefore, the result of Table 3 (b) strongly supports that the preference information is distilled to the student TOIST.

A simple taxonomy differs three scenarios where preference distillation works. As shown in Fig.5, the predicted results (filtered by a preference threshold of 0.9) of the TOIST models w/ or w/o preference distillation are compared. (1) Preference distillation makes the preference score of the false positive object (the baseball in the left picture) lower than the threshold. (2) The preference score of the false negative object (the spoon in the middle) is raised above the threshold with the distillation. (3) When there is no distillation, the false positive object (fork) scores higher than the true positive object (knife) (0.9822 > 0.9808). Although the distillation fails to lower the preference score of the false positive object below the threshold, its score is updated to be lower than the true positive one (0.9495 < 0.9680). These specific results demonstrate that the information of noun referring expression is distilled to the noun-agnostic student model in the form of preference scores.

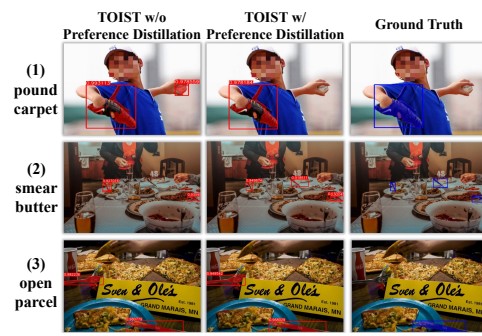

Figure 5: Examples of three scenarios where preference distillation clearly works.

## 5.5 Ablation Study and Qualitative Results

**Distillation Methods.** Instead of minimizing the distance between $l^{tr}_{pron}$ and $l^j_{c_s}$, a straightforward way is to directly minimize the distance between $l^{tr}_{pron}$ and $l^{tr}_{noun}$. As shown in Table 4, this simplified method does not work well, which prompts us to develop the distillation framework.



Table 4: Different distillation methods.

| Method | $mAP^{box}$ | $mAP^{mask}$ |
|---|---|---|
| TOIST | 41.3 | 35.2 |
| distill from $l^j_{c_s}$ to $l^{tr}_{pron}$ | **44.1** (+2.8) | **39.0** (+3.8) |
| distill from $l^{tr}_{noun}$ to $l^{tr}_{pron}$ | 41.9 (+0.6) | 36.0 (+0.8) |

Table 5: Results without pre-training.

| Method | $mAP^{box}$ | $mAP^{mask}$ |
|---|---|---|
| verb-pronoun input | 3.65 | 5.74 |
| verb-noun input | 11.19 | 12.67 |
| noun-pronoun distillation | 7.43 (+3.78) | 11.28 (+5.54) |



**Interaction of the Two Distillation Components.** In Table 3 (d) and (f), we show the effects of cluster loss or cluster center replacement together with soft binary target loss. (d) achieves +2.5% $mAP^{box}$ and +3.4% $mAP^{mask}$ improvement, which demonstrates the two distillation losses collaborate well. (f) only achieves +1.0% $mAP^{box}$ and +2.1% $mAP^{mask}$ improvement, slightly higher than (e) (using cluster loss only) but lower than (b) (using soft binary target loss only). This shows that preference distillation effectively improves object preference modeling. But solely replacing pronoun features to indicate target objects weakens the effect of preference distillation.

**Ablations for Cluster Number** K. Fig.3 (b) shows the ablations for cluster number K. We perform distillation experiments on different K values between 1 to 10 because increasing K to an even higher value makes the clustering task more difficult. All of the experiments yield better results than the plain TOIST (41.3% $mAP^{box}$, 35.2% $mAP^{mask}$) and K = 3 works the best. This demonstrates that a modest K can better cluster the information of noun features and distill it to the student TOIST.

Table 6: Ablations for pronoun input.

| Method | Pronoun | $mAP^{box}$ | $mAP^{mask}$ |
|---|---|---|---|
| TOIST | something | 41.3 | 35.2 |
| | it | 41.3 | 35.2 |
| | them | 41.4 | 35.0 |
| | abcd | 39.0 | 33.2 |
| TOIST w/ distillation | something | 44.1 | 39.0 |
| | it | 43.8 | 38.4 |
| | them | 43.8 | 38.1 |
| | abcd | 42.8 | 37.4 |

**Ablations for Pronoun Input.** Table 6 shows the results of TOIST with different pronoun input. In the plain TOIST and TOIST with distillation, the usage of *something*, *it* or *them* leads to similar results,

Table 7: Ablations for the task number $n_{\text{task}}$ on task oriented object detection.

| Task
Method | step on
something | sit
comfortably | place
flowers | get potatoes
out of fire | water
plant |
|---|---|---|---|---|---|
| TOIST w/o dis | 44.0 | 39.5 | 46.7 | 43.1 | 53.6 |
| dis $n_{\text{task}} = 14$ | 46.2 (+2.2) | 39.6 (+0.1) | 49.9 (+3.2) | 47.1 (+4.0) | 54.5 (+0.9) |
| dis $n_{\text{task}} = 5$ | 46.4 (+2.4) | 40.7 (+1.2) | **51.3 (+4.6)** | 46.8 (+3.7) | 54.6 (+1.0) |
| dis $n_{\text{task}} = 1$ | **47.0 (+3.0)** | **42.1 (+2.6)** | 50.8 (+4.1) | **47.4 (+4.3)** | **55.2 (+1.6)** |

while a meaningless string *abcd* yields less improvement. Nevertheless, the proposed distillation framework can still work well in the last case, which demonstrates the robustness of our method.

**Results without Pre-training.** In our architecture, the pre-trained noun referring expression comprehension models are leveraged. To investigate whether the noun-pronoun distillation framework is a standalone technical contribution, we conduct experiments without pre-training. The models are trained from scratch on the COCO-Tasks dataset and the results are shown in Table 5, which demonstrates that the proposed distillation can still improve performance even without pre-training.

**Ablations for Task Number.** Table 7 shows the ablation study of different task numbers, in which the first row corresponds to the plain TOIST without distillation and the others show the results with distillation under different $n_{\text{task}}$. The results demonstrate our proposed distillation works for different $n_{\text{task}}$, even if $n_{\text{task}} = 1$. And overall, smaller $n_{\text{task}}$ leads to better performance. We attribute this to the reduced problem complexity due to the less interaction between different tasks, which makes it easier to improve the ability of the model to understand verbs through noun-pronoun distillation.

**Qualitative Results.** Fig.6 shows more qualitative results. In (a), two toilets are taken as target objects and annotated partially or totally. But TOIST simultaneously predicts the two kinds of results for each toilet. In (b), no object is annotated, while TOIST keenly detects two water bottles that afford the task. In (c), TOIST predicts more accurate mask result than ground truth. In (d), the table is selected and interestingly it does afford the task as the table edge can be used to open beer bottles. More qualitative results can be found in Appendix E.

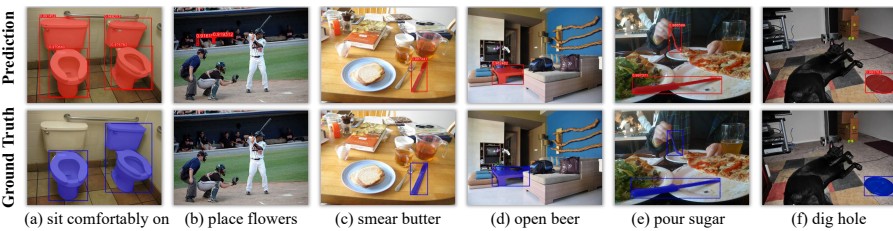

Figure 6: Qualitative results of proposed method for task oriented instance segmentation.

## 6 Conclusion and Discussion

We explore the problem of task oriented instance segmentation and propose a transformer-based method named as TOIST with a novel noun-pronoun distillation framework. Experiments show our method successfully models affordance and preference, achieving SOTA results on the COCO-Tasks dataset. **Limitations.** Due to the lack of large-scale datasets with more abundant tasks, TOIST is only evaluated on limited tasks. While this is sufficient for many robotics applications, it would be interesting to explore general verb reference understanding on more tasks. **Potential Negative Social Impact.** Because TOIST is not perfect, when it is used in robotics applications, robots may have difficulty in selecting the most suitable object to carry out a task or even cause damage.

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
