# TOIST: Task Oriented Instance Segmentation Transformer with Noun-Pronoun Distillation Supplementary Material

**Pengfei Li[1]   Beiwen Tian[1]   Yongliang Shi[1]   Xiaoxue Chen[1]**
**Hao Zhao[2,3]   Guyue Zhou[1]   Ya-Qin Zhang[1]**
[1]AIR, Tsinghua University    [2] Peking University    [3]Intel Labs
li-pf22@mails.tsinghua.edu.cn
zhao-hao@pku.edu   hao.zhao@intel.com

https://github.com/AIR-DISCOVER/TOIST

## 1   Implementation Details

### 1.1   Method

**Noun Features.**   As mentioned in Section 3 (formulation) of the main paper, in an input image, it is possible that no objects or multiple objects afford a specific task. And in the latter case, the objects may belong to multiple classes. But for the language input $\mathbf{X}_l$ of verb-noun form, the noun corresponds to the ground truth object categories. Therefore, when the count of targets $n_{gt} = 0$, we use an empty string to construct $\mathbf{X}_l$. When $n_{gt} > 0$ and all target objects belong to the same category, we take a phrase like *sit comfortably on sofa* as $\mathbf{X}_l$. When $n_{gt} > 0$ and the target objects belong to multiple classes, the language input $\mathbf{X}_l$ is set to the concatenation of multiple phrasal verbs, such as *sit comfortably on chair sit comfortably on bed*.

We only update the proposed memory bank in the latter two cases. In these cases, if a noun is encoded into multiple tokenized features by the text encoder, we use the mean value of the features processed by the transformer encoder as $l_{noun}^{tr}$ for updating. In the last case, we take the average of multiple noun features $l_{noun-1}^{tr}, \ldots, l_{noun-n_c}^{tr}$ as the final $l_{noun}^{tr}$, where $n_c$ is the count of classes. In this way, the privileged knowledge of multiple nouns is more easily distilled into the single pronoun feature of the student model when an image contains multiple classes of objects equally suitable for a task.

**Components of TOIST.**   RoBERTa-base [8] and ResNet-101 [3] are used as the text encoder and the CNN-based backbone (the image encoder). For the logit head and the detect head, two feed forward networks of depth one and three are leveraged, respectively. For the segment head, following the network design of [2], a multi-head attention operator and a FPN [5]-like convolutional neural network are used. After each block in the transformer decoder, TOIST generates auxiliary outputs [1] with the prediction heads.

### 1.2   Loss Functions

We present the loss terms used for the plain TOIST training in details.

For the ground truth objects $\mathbf{O}_{gt}$ of each training sample, we define $\mathbf{p}_i^{span} = [p_{i,1}^{span}, \ldots, p_{i,n_{max}}^{span}] \in [0,1]^{n_{max}}$ as a uniform distribution over the span positions of the text tokens corresponding to the $i$-th ground truth object. $p_{i,n_{max}}^{span}$ stands for the probability of "no-object", which is 0 for the ground truth objects. As a reminder, we use the whole verb-pronoun (or verb-noun) description as token span. Assuming that $n_{gt}$ (the count of $\mathbf{O}_{gt}$) is smaller that $n_{pred}$ (the count of the predicted objects $\mathbf{O}_{pred}$),

36th Conference on Neural Information Processing Systems (NeurIPS 2022).

we pad $\mathbf{O}_{\text{gt}}$ with $\varnothing$ ("no-object") to be of size $n_{\text{pred}}$, denoted as $\mathbf{O}'_{\text{gt}}$. For $\varnothing$, $\mathbf{p}_i^{\text{span}}$ is assigned to be $p_{i,j}^{\text{span}} = \mathbb{1}_{\{j=n_{\text{max}}\}}$: if $j = n_{\text{max}}$, $p_{i,j}^{\text{span}} = 1$; otherwise, $p_{i,j}^{\text{span}} = 0$.

We denote the bipartite matching between $\mathbf{O}'_{\text{gt}}$ and $\mathbf{O}_{\text{pred}}$ as $\hat{\sigma}_0$, which is calculated by minimizing the matching loss with the Hungarian algorithm [2]:

$$\hat{\sigma}_0 = \underset{\sigma_0 \in \mathfrak{S}_{n_{\text{pred}}}}{\arg\min} \sum_{i}^{n_{\text{pred}}} \mathbb{1}_{\{p_{i,n_{\text{max}}}^{\text{span}}=0\}} \left[ \mathcal{L}_{\text{l1}}(b_i, \hat{b}_{\sigma_0(i)}) + \mathcal{L}_{\text{giou}}(b_i, \hat{b}_{\sigma_0(i)}) + \mathcal{L}_{\text{token-m}}(\mathbf{p}_i^{\text{span}}, \hat{\mathbf{g}}_{\sigma_0(i)}) \right].$$
(1)

Here, $\mathfrak{S}_{n_{\text{pred}}}$ is the set of all permutations of $n_{\text{pred}}$ elements. $b_i$ and $\hat{b}_{\sigma_0(i)}$ are the ground truth box and the predicted box, respectively. $\hat{\mathbf{g}}_{\sigma_0(i)}$ is the predicted logit, as detailed in the main paper. The loss terms are defined as follows:

$$\mathcal{L}_{\text{l1}}(b_i, \hat{b}_{\sigma_0(i)}) = \left\| b_i - \hat{b}_{\sigma_0(i)} \right\|_1.$$
(2)

$$\mathcal{L}_{\text{giou}}(b_i, \hat{b}_{\sigma_0(i)}) = 1 - \left( \frac{|b_i \cap \hat{b}_{\sigma_0(i)}|}{|b_i \cup \hat{b}_{\sigma_0(i)}|} - \frac{|B(b_i, \hat{b}_{\sigma_0(i)}) \backslash b_i \cup \hat{b}_{\sigma_0(i)}|}{|B(b_i, \hat{b}_{\sigma_0(i)})|} \right).$$
(3)

$$\mathcal{L}_{\text{token-m}}(\mathbf{p}_i^{\text{span}}, \hat{\mathbf{g}}_{\sigma_0(i)}) = - \sum_{j}^{n_{\text{max}}} p_{i,j}^{\text{span}} \frac{\exp\left(\hat{g}_j^{\sigma_0(i)}\right)}{\sum_{l=1}^{n_{\text{max}}} \exp\left(\hat{g}_l^{\sigma_0(i)}\right)}.$$
(4)

$\mathcal{L}_{\text{giou}}$ is the Generalized Intersection over Union (GIoU) loss [11]. $|\cdot|$ calculates the size of an area. $B(b_i, \hat{b}_{\sigma_0(i)})$ stands for the smallest box containing $b_i$ and $\hat{b}_{\sigma_0(i)}$. The calculation of $\mathcal{L}_{\text{giou}}$ is implemented by linear functions, so it is differentiable and can be used for back propagation.

For segmentation, Dice/F-1 loss [9] $\mathcal{L}_{\text{dice}}$ and Focal cross-entropy loss [6] $\mathcal{L}_{\text{cross}}$ are leveraged:

$$\mathcal{L}_{\text{dice}}(m_i, \hat{m}_{\sigma_0(i)}) = 1 - \frac{2m_i \delta(\hat{m}_{\sigma_0(i)}) + 1}{\delta(\hat{m}_{\sigma_0(i)}) + m_i + 1}.$$
(5)

Here, $m_i$ is the ground truth instance mask of the $i$-th object. $\hat{m}_{\sigma_0(i)}$ is the corresponding predicted mask logits. $\delta$ is the sigmoid function.

$$\mathcal{L}_{\text{cross}}(m_i, \hat{m}_{\sigma_0(i)}) = -\alpha_t (1 - p_t)^{\gamma} \left[ m_i \log \delta(\hat{m}_{\sigma_0(i)}) + (1 - m_i) \log(1 - \delta(\hat{m}_{\sigma_0(i)})) \right],$$
(6)

where

$$\alpha_t = \alpha m_i + (1 - \alpha)(1 - m_i),$$
(7)

$$p_t = m_i \delta(\hat{m}_{\sigma_0(i)}) + (1 - m_i)(1 - \delta(\hat{m}_{\sigma_0(i)})),$$
(8)

$\alpha$ and $\gamma$ are hyper-parameters.

The soft-token prediction loss $\mathcal{L}_{\text{token}}$ is defined as:

$$\mathcal{L}_{\text{token}}(\mathbf{p}_i^{\text{span}}, \hat{\mathbf{g}}_{\sigma_0(i)}) = - \sum_{j}^{n_{\text{max}}} p_{i,j}^{\text{span}} \log \frac{\exp\left(\hat{g}_j^{\sigma_0(i)}\right)}{\sum_{l=1}^{n_{\text{max}}} \exp\left(\hat{g}_l^{\sigma_0(i)}\right)}.$$
(9)

The contrastive alignment loss is used to align the embedded features of the predicted objects and the corresponding text tokens. The embedded features are obtained by projecting the processed text features of the transformer encoder and the output features of transformer decoder to the same smaller dimension. We follow the definition of [4]:

$$\mathcal{L}_{\text{align}} = \frac{1}{2} \sum_{i}^{n_{\text{pred}}} \frac{1}{|T_i^+|} \sum_{j \in T_i^+} - \log \frac{\exp\left(o_i^\top t_j / \tau\right)}{\sum_{k=1}^{n_{\text{max}}} \exp\left(o_i^\top t_k / \tau\right)}$$

$$+ \frac{1}{2} \sum_{i}^{n_{\text{max}}} \frac{1}{|O_i^+|} \sum_{j \in O_i^+} - \log \frac{\exp\left(t_i^\top o_j / \tau\right)}{\sum_{k=1}^{n_{\text{pred}}} \exp\left(t_i^\top o_k / \tau\right)}.$$
(10)

$T_i^+$ is the set of token features that a predicted object feature $o_i$ should be aligned to. $O_i^+$ is the set of object features to be aligned with a token feature $t_i$. Here the predicted objects matched to $\varnothing$ are not included. $\tau$ is a hyper-parameter.

Finally, the total loss for the plain TOIST can be written as:

$$
\begin{aligned}
\mathcal{L}_{\mathrm{TOIST}} = {}& \mathbb{1}_{\{p_{i,n_{\max}}^{\mathrm{span}}=0\}}[\lambda_1\mathcal{L}_{\mathrm{l1}}(b_i,\hat{b}_{\hat{\sigma_0}(i)}) + \lambda_2\mathcal{L}_{\mathrm{giou}}(b_i,\hat{b}_{\hat{\sigma_0}(i)})] \\
& + \mathbb{1}_{\{p_{i,n_{\max}}^{\mathrm{span}}=0\}}[\lambda_3\mathcal{L}_{\mathrm{dice}}(m_i,\hat{m}_{\hat{\sigma_0}(i)}) + \lambda_4\mathcal{L}_{\mathrm{cross}}(m_i,\hat{m}_{\hat{\sigma_0}(i)})] \\
& + \lambda_5\mathcal{L}_{\mathrm{token}}(\mathbf{p}_i^{\mathrm{span}},\hat{\mathbf{g}}_{\hat{\sigma_0}(i)}) \\
& + \lambda_6\mathcal{L}_{\mathrm{align}},
\end{aligned}
\tag{11}
$$

### 1.3 Hyper-parameters and Training Details

For the proposed architecture, we set $d = 256$, $n_{\max} = 256$, $n_{tr} = 6$, $n_{\mathrm{pred}} = 100$, $n_{\mathrm{mem}} = 1024$ and K $= 3$. For the loss functions, we set $\lambda_1 = 5$, $\lambda_2 = 2$, $\lambda_3 = 1$, $\lambda_4 = 1$, $\lambda_5 = 1$, $\lambda_6 = 1$, $\lambda_7 = 10^4$, $\lambda_8 = 50$, $\alpha = 0.25$, $\gamma = 2$ and $\tau = 0.07$.

During training, we augment input images with random resize and random crop. Specifically, each image is resized such that the shortest side is between 480 and 800 pixels and the longest side is less than 1333 pixels. With probability 0.5, an image is cropped to a random size, where each side is between 384 and 1333 pixels.

We implement TOIST with PyTorch [10]. Both of the student and teacher TOIST models are initialized with the model pre-trained by [4]. We fine-tune the two models on the COCO-Tasks dataset separately for 30 epochs. Then we use the fine-tuned teacher model to distill knowledge to the student model for 15 epochs. We train the models with an AdamW optimizer. The initial learning rates are set to $10^{-5}$, $10^{-5}$, $5 \times 10^{-5}$ for text encoder, backbone and transformer, respectively. The weight decay is $10^{-4}$. Our experiments were preformed on 8 NVIDIA A100 GPUs.

## 2 Dataset Details

We perform experiments on the COCO-Tasks dataset [12] which re-annotates the COCO dataset [7] with preference-aware affordance labels. The COCO-Tasks dataset contains 14 tasks. For each task, there are 3600 train images and 900 test images. We train the proposed architecture on all the train images and evaluate it on all the test images.

In an image, the most suitable objects (one or more) for solving the task are selected and their bounding boxes are taken as ground truth labels for detection. The number of selected objects in an image varies from zero to a dozen. For each task, the total number of selected objects varies between 1,105 and 9,870 and the number of different object categories varies between 6 and 30. Totally, the COCO-Tasks dataset contains 65797 selected objects spanning 49 of the 80 COCO object class categories. This shows the diversity of the dataset. For each task, the total count of all instances belonging to the selected categories largely varies between 7,172 and 34,160. This shows the task oriented object detection problem on this dataset is a non-trivial problem and solving it with traditional methods is very challenging. The preference between multiple classes and multiple instances of the same class must be taken into account.

The COCO-Tasks dataset is annotated with the available COCO detection boxes. Leveraging the corresponding COCO segmentation masks directly gives the upgraded instance segmentation version. We evaluate our proposed method on the upgraded dataset.

## 3 Quantitative results

**Strategies for Updating Memory Bank.** To update the text feature memory bank in noun-pronoun distillation, two strategies are compared: (1) First-in-first-out. (2) Replacing the feature closest to the new-coming $l_{\mathrm{noun}}^{\mathrm{tr}}$. As shown in Table 1, the second strategy leads to better performance.

**Comparison to the Baseline with the Same Backbone.** To investigate whether our TOIST architecture is a standalone technical contribution by marginalizing the benefits brought by pre-trained models, we present another baseline 'MDETR+GGNN'. Specially, to leverage the knowledge in noun

Table 1: Comparison of different updating strategies for the memory bank in the distillation.

| Method | mAP$^{\text{box}}$ | mAP$^{\text{mask}}$ |
|---|---|---|
| TOIST w/o distillation | 41.3 | 35.2 |
| first-in-first-out | 43.9 (+2.6) | 38.8 (+3.6) |
| replacing the closest one | **44.1 (+2.8)** | **39.0 (+3.8)** |

Table 2: Comparison of the proposed method to 'MDETR+GGNN' baseline on the COCO-Tasks dataset.

| Method | mAP$^{\text{box}}$ | mAP$^{\text{mask}}$ |
|---|---|---|
| MDETR + GGNN w/o pretraining | 9.6 | 8.6 |
| MDETR + GGNN | 36.8 | 30.3 |
| TOIST | 41.3 (+4.5) | 35.2 (+4.9) |
| TOIST w/ distillation | **44.1 (+7.3)** | **39.0 (+8.7)** |

referring expression comprehension, we use the official pre-trained model of MDETR [4] and then fine-tune it on the COCO-Task dataset. We use the class names of the ground truth objects in each image as the text input to detect these objects. Then we use the GGNN model [12] to infer which objects are preferred for a task. The results are shown in Table 2. Note that this baseline is also tested with privileged noun ground truth, but our distillation method only use the privileged knowledge during training. Nevertheless, our proposed method still has a significant performance improvement over this strong baseline (+7.3% mAP$^{\text{box}}$ and +8.7% mAP$^{\text{mask}}$). This demonstrates that our TOIST architecture is a standalone technical contribution towards task oriented instance segmentation and pretraining is necessary but insufficient to get the performance level of TOIST.

**Ablations for Loss Terms.** To demonstrate the effectiveness of the used loss terms, we provide ablation studies in which we remove $\mathcal{L}_{\text{align}}$ or $\mathcal{L}_{\text{token}}$ or both. The quantitative results are demonstrated in Table 3. It shows that removing $\mathcal{L}_{\text{token}}$ brings a performance drop of -1.2% mAP$^{\text{box}}$ and -0.4% mAP$^{\text{mask}}$, because the association between the matched object predictions and the task descriptions is weakened. Removing $\mathcal{L}_{\text{align}}$ brings a performance drop of -0.2% mAP$^{\text{box}}$ and -0.1% mAP$^{\text{mask}}$, because the features of an object and its corresponding text features cannot be explicitly constrained to be closer. Interestingly, removing both of them brings a significant performance drop of -17.9% mAP$^{\text{box}}$ and -14.5% mAP$^{\text{mask}}$, implying the two loss terms enhance the effect of each other to make TOIST understand verb reference better.

**Replicates.** In Table 4 and 5, we show the mean and standard deviation of the results obtained by running experiments three times under different random seeds.

**Per-task Results.** In Table 6 and 7, we provide per-task results of the proposed method and existing state-of-the-art baselines on the COCO-Tasks dataset. The results show that our TOIST model with noun-pronoun distillation achieves the best performance in most tasks.

## 4 More Analysis

In Fig.1, we present the percentage of ground truth object categories, in each task. The figures show that the distribution of object categories in each task in the COCO-Tasks dataset is very diverse. In

Table 3: Ablations for the soft-token prediction loss and the contrastive alignment loss.

| Method | mAP$^{\text{box}}$ | mAP$^{\text{mask}}$ |
|---|---|---|
| TOIST | **41.3** | **35.2** |
| TOIST w/o $\mathcal{L}_{\text{token}}$ | 40.1 (-1.2) | 34.8 (-0.4) |
| TOIST w/o $\mathcal{L}_{\text{align}}$ | 41.1 (-0.2) | 35.1 (-0.1) |
| TOIST w/o $\mathcal{L}_{\text{token}}$ and $\mathcal{L}_{\text{align}}$ | 23.4 (-17.9) | 20.7 (-14.5) |

Table 4: Ablations for distillation settings. CCR, CL and SBTL are short for cluster center replacement, cluster loss and soft binary target loss, respectively.

| Index | CCR | CL | SBTL | mAP$^{box}$ | mAP$^{mask}$ |
|-------|-----|-----|------|-------------|--------------|
| (a) | × | × | × | 41.3±0.44 | 35.0±0.28 |
| (b) | × | × | ✓ | 43.2±0.20 | 38.0±0.02 |
| (c) | × | ✓ | × | 42.0±0.03 | 37.1±0.04 |
| (d) | × | ✓ | ✓ | 43.5±0.30 | 38.6±0.05 |
| (e) | ✓ | × | × | 42.0±0.03 | 36.9±0.06 |
| (f) | ✓ | × | ✓ | 42.3±0.02 | 37.3±0.02 |
| (g) | ✓ | ✓ | × | 42.3±0.02 | 37.5±0.03 |
| (h) | ✓ | ✓ | ✓ | **44.1±0.12** | **39.0±0.07** |

Table 5: Ablations for pronoun input.

| Method | Pronoun | mAP$^{box}$ | mAP$^{mask}$ |
|--------|---------|-------------|--------------|
| TOIST | something | 41.3±0.44 | 35.0±0.28 |
| | it | 41.4±0.26 | 35.1±0.20 |
| | them | 41.5±0.28 | 34.7±0.28 |
| | abcd | 38.9±0.10 | 33.2±0.15 |
| TOIST w/ distillation | something | 44.1±0.12 | 39.0±0.07 |
| | it | 43.8±0.12 | 38.4±0.02 |
| | them | 43.7±0.13 | 38.1±0.03 |
| | abcd | 42.8±0.06 | 37.4±0.02 |

Table 6: Per-task object detection results on COCO-Tasks.

| | Object Detection, AP@0.5 | | | | | | | | | | | | | | |
|--------|------|------|------|------|------|------|------|------|------|------|------|------|------|------|------|
| Method | 1 | 2 | 3 | 4 | 5 | 6 | 7 | 8 | 9 | 10 | 11 | 12 | 13 | 14 | mean |
| Faster-RCNN | 28.1 | 25.8 | 30.1 | 22.0 | 30.5 | 11.7 | 30.8 | 0.0 | 5.1 | 33.4 | 9.7 | 6.1 | 24.6 | 30.9 | 20.6 |
| Faster-RCNN + pick best | 22.9 | 18.1 | 19.8 | 15.0 | 21.3 | 5.8 | 20.4 | 3.9 | 3.3 | 22.0 | 11.1 | 5.0 | 12.5 | 15.6 | 14.1 |
| Faster-RCNN + ranker | 10.7 | 10.4 | 11.5 | 11.6 | 11.8 | 3.3 | 15.0 | 2.4 | 4.6 | 10.5 | 5.2 | 5.0 | 8.3 | 17.2 | 9.1 |
| Faster-RCNN + classifier | 33.1 | 26.7 | 36.8 | 32.9 | 35.4 | 14.6 | 40.3 | 14.4 | 17.6 | 38.4 | 17.1 | 24.5 | 33.2 | 38.1 | 28.8 |
| Faster-RCNN + GGNN | 36.6 | 29.8 | 40.5 | 37.6 | 41.0 | 17.2 | 43.6 | 17.9 | 21.0 | 40.6 | 22.3 | 28.4 | 39.1 | 40.7 | 32.6 |
| Yolo + GGNN | 36.8 | 31.9 | 39.1 | 38.0 | 41.6 | 16.5 | 44.4 | 18.7 | 23.0 | 39.0 | 22.3 | 26.9 | 44.0 | 42.0 | 33.2 |
| Mask-RCNN | 30.8 | 25.0 | 32.7 | 20.4 | 33.1 | 8.0 | 28.2 | 7.9 | 11.2 | 43.1 | 8.1 | 14.7 | 32.2 | 32.0 | 23.4 |
| Mask-RCNN + pick best | 21.9 | 19.6 | 22.1 | 21.6 | 28.3 | 12.6 | 26.2 | 3.4 | 3.6 | 29.1 | 20.4 | 3.7 | 22.4 | 28.6 | 18.8 |
| Mask-RCNN + ranker | 11.6 | 11.6 | 12.2 | 14.0 | 15.0 | 5.0 | 18.9 | 2.6 | 3.3 | 12.2 | 7.1 | 4.7 | 9.2 | 21.3 | 10.6 |
| Mask-RCNN + classifier | 36.8 | 31.1 | 42.4 | 39.5 | 40.8 | 18.6 | 48.3 | 13.9 | 17.2 | 48.4 | 21.4 | 23.1 | 43.5 | 46.4 | 33.7 |
| Mask-RCNN + GGNN | 39.0 | 33.2 | 46.4 | _43.8_ | 47.7 | 21.4 | 51.2 | 16.7 | 20.3 | **51.3** | 27.3 | 26.8 | _50.2_ | 48.9 | 37.4 |
| MDETR + GGNN | _44.3_ | 36.5 | 45.2 | _28.6_ | 44.0 | **27.6** | 35.9 | 20.7 | **34.7** | 46.3 | 27.8 | 41.5 | _46.5_ | 36.2 | 36.8 |
| TOIST | 44.0 | _39.5_ | _46.7_ | 43.1 | **53.6** | 23.5 | _52.8_ | _21.3_ | 32.0 | 46.3 | _33.1_ | _41.7_ | 48.1 | _52.9_ | _41.3_ |
| TOIST w/ distillation | **45.8** | **40.0** | **49.4** | **49.6** | _53.4_ | _26.9_ | **58.3** | **22.6** | _32.5_ | _50.0_ | **35.5** | **43.7** | 52.8 | **56.2** | **44.1** |

Fig.2, we present the percentage of the images that contain at least one ground truth object or contain no object in each task. These figures show that many of the images do not contain objects that are suitable for the tasks. All these statistical results demonstrate that it is non-trivial to find the objects with affordance and preference for a specific task in a given image.

We present the precision-recall curves of our detection and segmentation results on all the test data or the test data that contain at least one ground truth object (termed nonempty data) in Fig.3-6. It can be seen that the curves of the results with noun input are generally higher than those with pronoun input. The proposed noun-pronoun distillation makes the curves of the results with pronoun input closer to those with noun input, which demonstrates the effectiveness of the distillation method. Meanwhile, on the nonempty data, the noun-pronoun distillation only marginally improves TOIST, indicating that the method makes TOIST more capable of filtering out objects that do not afford the tasks in the empty data.

We further present the precision-recall curves of our results on the test data that contain objects of specified classes in each task. The results show that the distillation method has different performance on different categories. For example, in the task *step on something*, it works for the data containing

Table 7: Per-task instance segmentation results on COCO-Tasks.

| | Instance Segmentation, AP@0.5 | | | | | | | | | | | | | | |
|--------|------|------|------|------|------|------|------|------|------|------|------|------|------|------|------|
| Method | 1 | 2 | 3 | 4 | 5 | 6 | 7 | 8 | 9 | 10 | 11 | 12 | 13 | 14 | mean |
| Mask-RCNN | 23.8 | 21.0 | 33.0 | 12.1 | 33.0 | 6.8 | 19.3 | 6.4 | 7.7 | 43.0 | 6.2 | 10.9 | 31.8 | 24.8 | 20.0 |
| Mask-RCNN + pick best | 17.9 | 16.4 | 21.7 | 16.8 | 27.9 | 10.6 | 23.3 | 2.9 | 2.8 | 28.7 | 18.6 | 2.3 | 21.9 | 23.4 | 16.8 |
| Mask-RCNN + ranker | 9.0 | 10.0 | 11.9 | 11.2 | 15.3 | 4.4 | 14.4 | 1.8 | 2.3 | 12.0 | 6.2 | 3.1 | 9.1 | 19.1 | 9.3 |
| Mask-RCNN + classifier | 30.0 | 27.3 | 41.4 | 30.3 | 39.9 | 14.8 | 35.7 | 12.5 | 13.5 | 46.8 | 19.4 | 16.0 | 43.0 | 38.1 | 29.2 |
| Mask-RCNN + GGNN | 31.8 | 28.6 | _45.4_ | 33.7 | 46.8 | 16.6 | 37.8 | 15.1 | 15.0 | **49.9** | 24.9 | 18.9 | _49.8_ | 39.7 | 32.4 |
| MDETR + GGNN | 36.9 | 31.3 | _43.6_ | 17.1 | 42.9 | _20.1_ | 19.9 | _18.7_ | _24.5_ | 45.5 | 23.1 | _30.9_ | _46.2_ | 24.0 | 30.3 |
| TOIST | _37.0_ | _34.4_ | 44.7 | _34.2_ | _51.3_ | 18.6 | _40.5_ | 17.1 | 23.4 | 43.8 | _29.3_ | 29.9 | 46.6 | _42.4_ | _35.2_ |
| TOIST w/ distillation | **40.8** | **36.5** | **48.9** | **37.8** | **53.4** | **22.1** | **44.4** | **20.3** | **26.9** | _48.1_ | **31.8** | **34.8** | **51.5** | **46.3** | **38.8** |

*chair* but does not work for those containing *dining table* or *couch* (see the first row of Fig.7). However, in the task *place flowers* and *get potatoes out of fire*, the proposed distillation method works for almost every category (see the fifth row to the twelfth row of Fig.7). Combined with the statistics of category proportion in Fig.1, we think one possible reason is that the effect of the distillation on different categories is influenced by the proportion of categories in the tasks. When a few classes take a large portion of selected objects in a certain task, the effect of the distillation on these classes is good, while that on others is poor. If the number of categories in the whole task is distributed more evenly, the distillation can boost performance for most categories.

## 5   More Qualitative results

We present more qualitative results in Fig.9. For each task, we select four diverse scenes for visualization. These results show that our method is robust to complex scenes of different tasks.

## 6   Used and Released Asserts

The license of the assets we used is as follows: (a) MIT License for the COCO-Tasks dataset. (b) Creative Commons Attribution 4.0 License for the Microsoft COCO dataset. (c) Apache License 2.0 for MDETR, DETR and Mask-RCNN implemented by Detectron2. All existing codes and dataset we used are open-source and allowed for research. To avoid the disclosure of personally identifiable information and the presentation of the content that might be considered offensive, we have blurred out some of the figures in this paper. Our code is released under the MIT license.

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

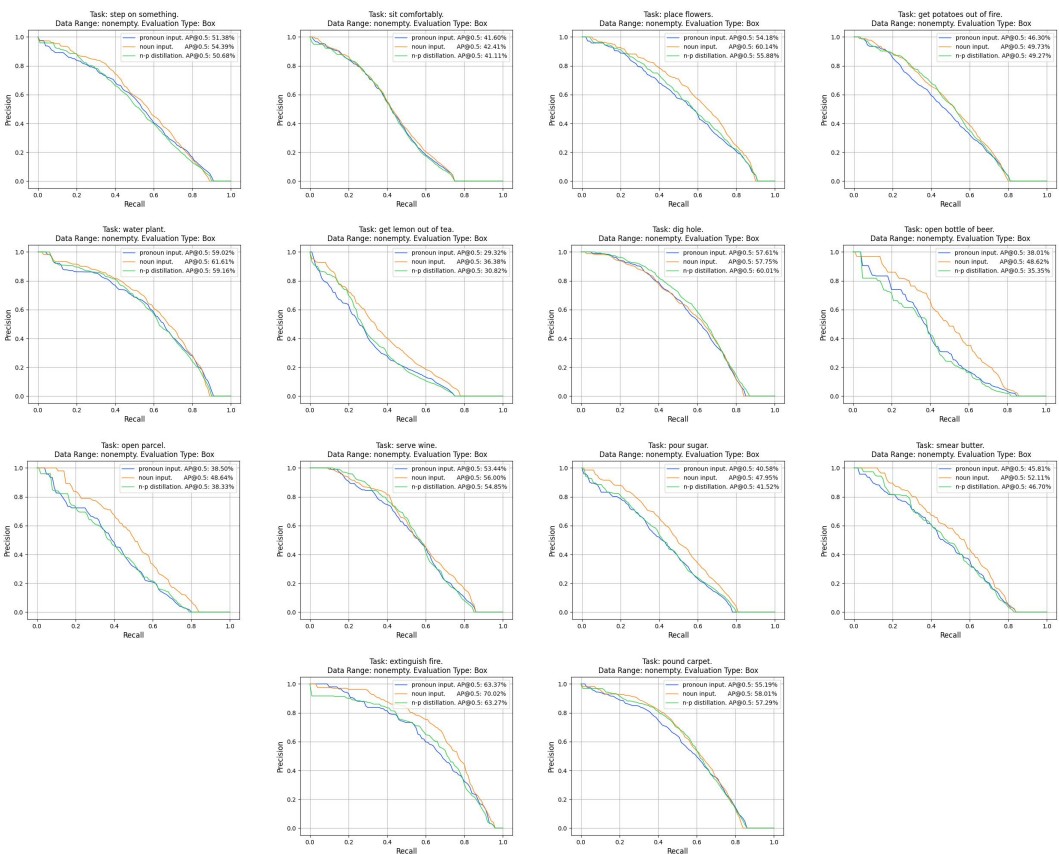

Figure 4: The precision-recall curves for object detection on the test data that contain selected objects in each task. The evaluation type 'Box' means object box instead of object mask.

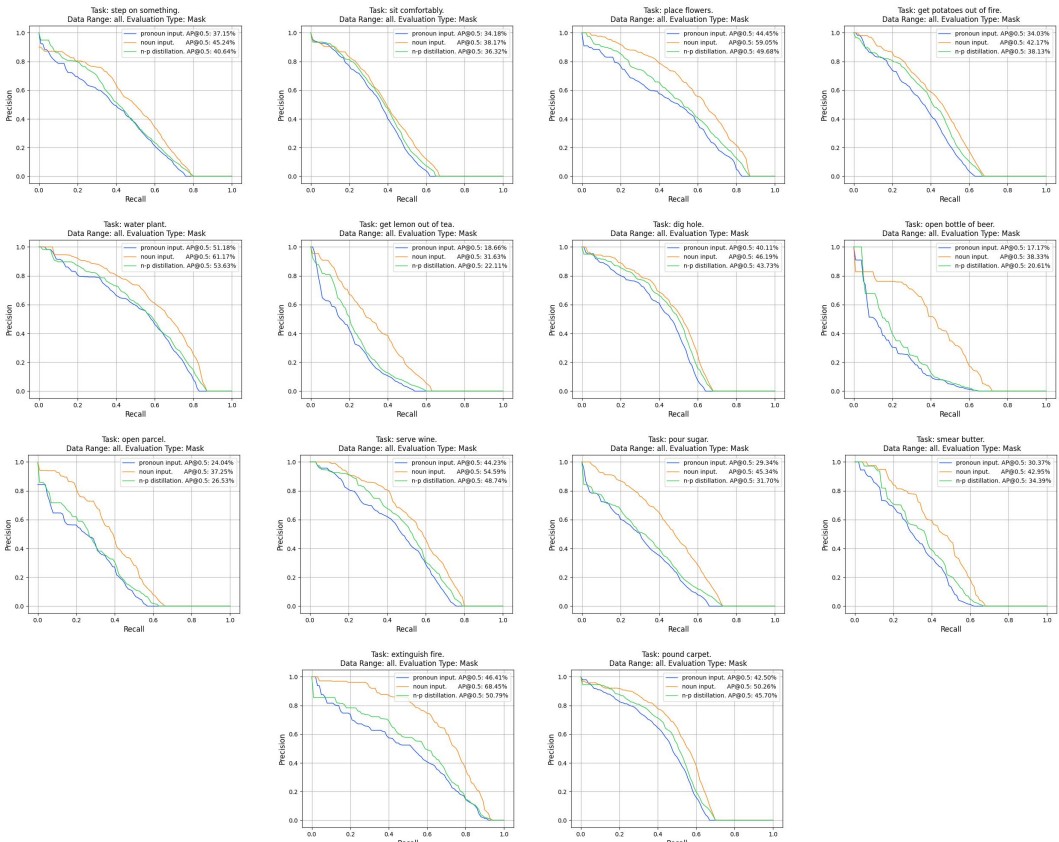

Figure 5: The precision-recall curves for instance segmentation on all the test data in each task. The evaluation type 'Mask' means object mask instead of object box.

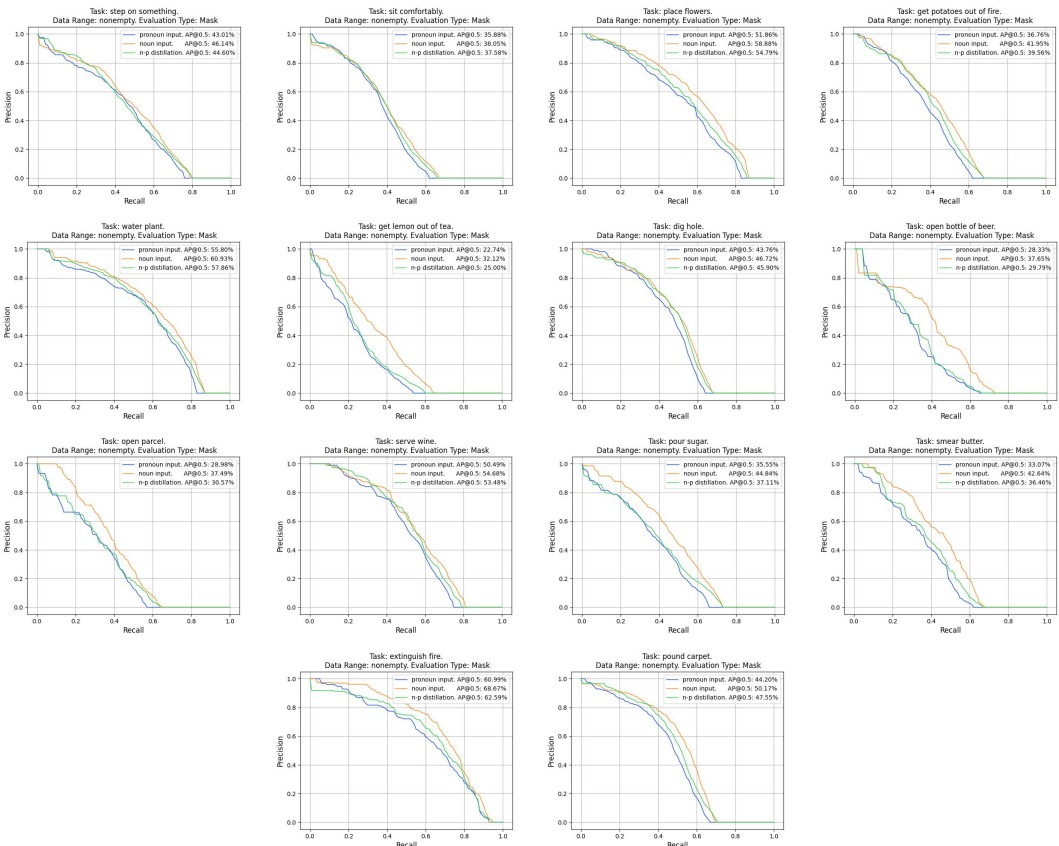

Figure 6: The precision-recall curves for instance segmentation on the test data that contain selected objects in each task. The evaluation type 'Mask' means object mask instead of object box.

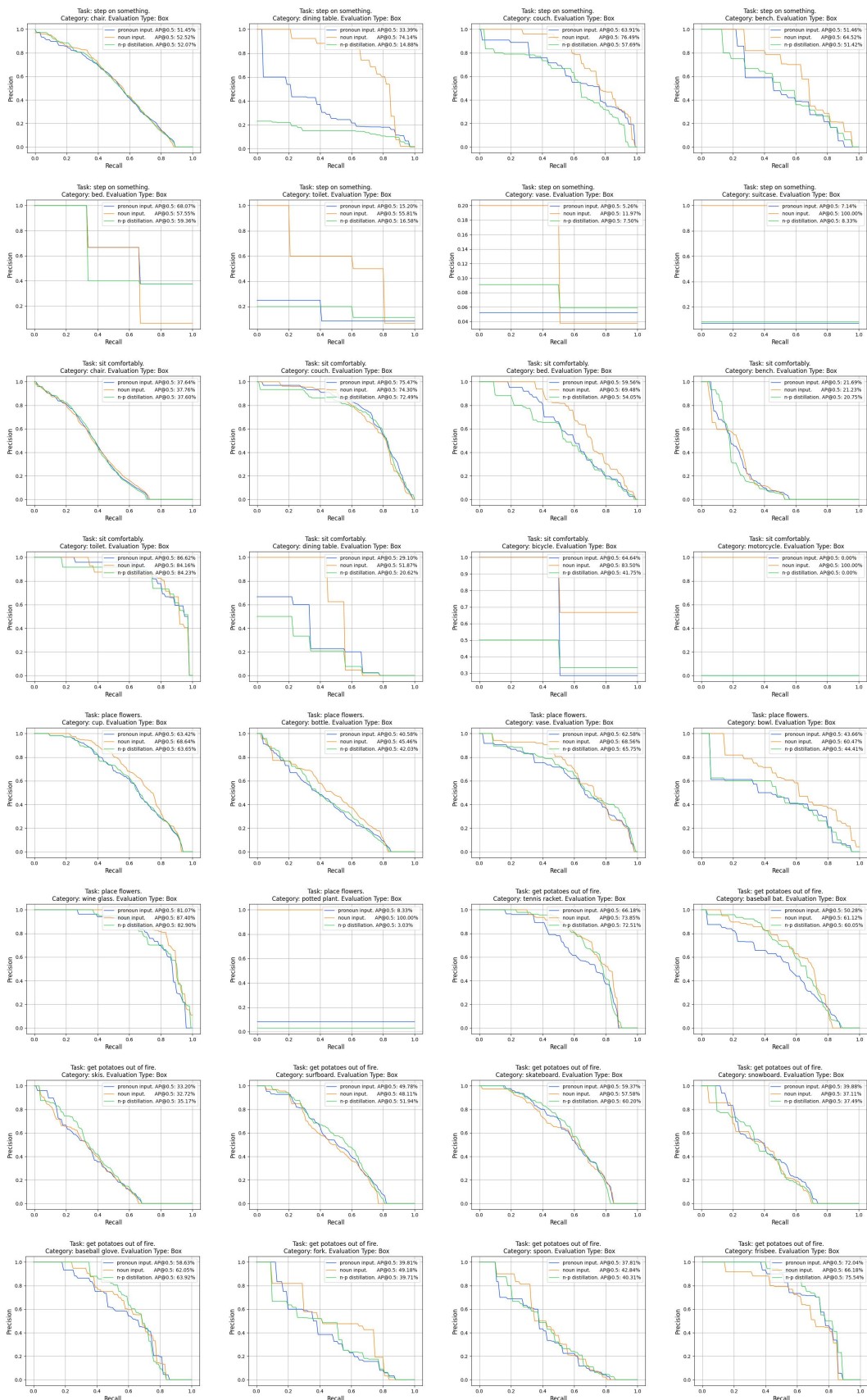

Figure 7: The precision-recall curves for object detection on the test data that contain objects of specified classes in each task.

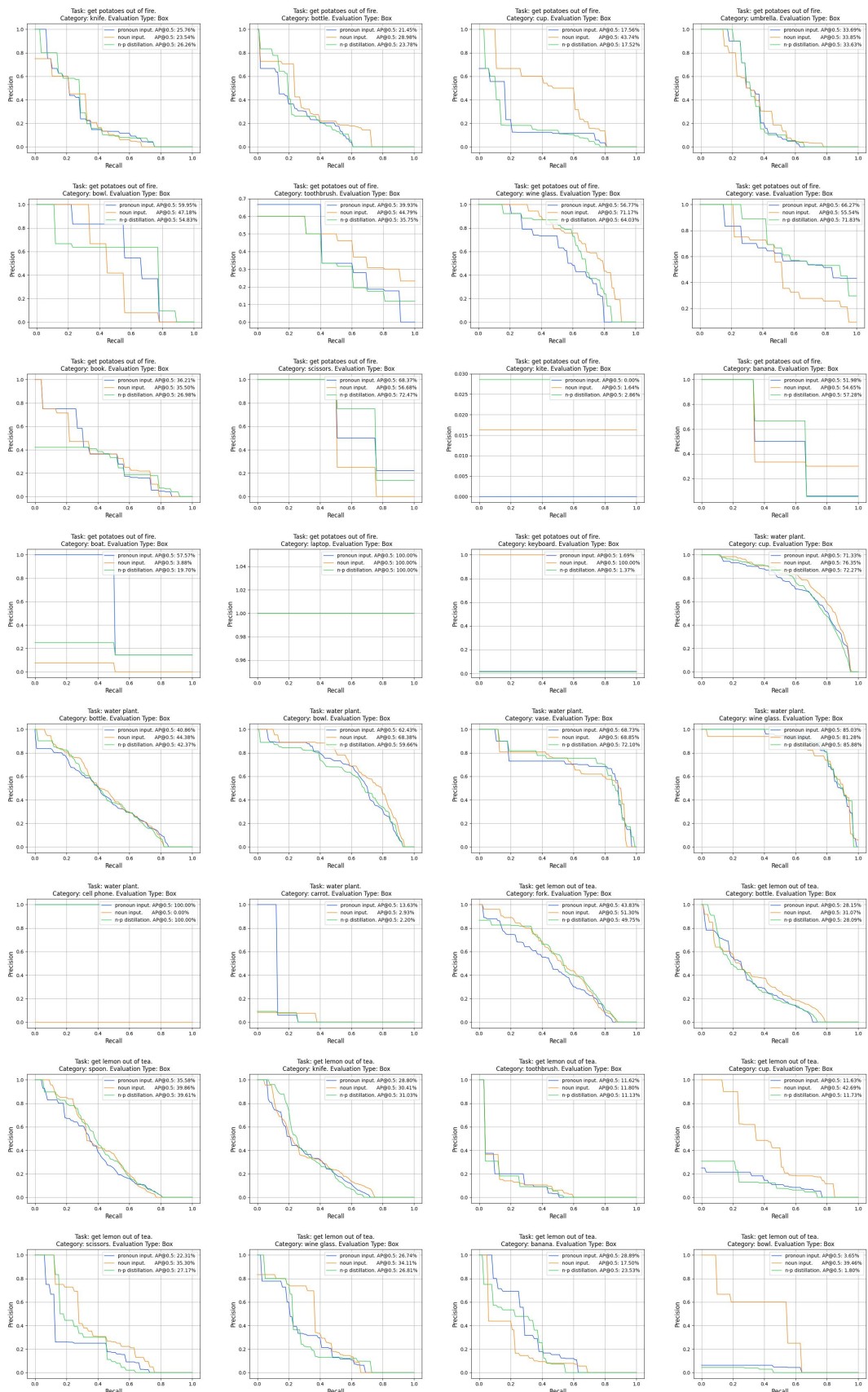

Figure 7: The precision-recall curves for object detection on the test data that contain objects of specified classes in each task (cont.).

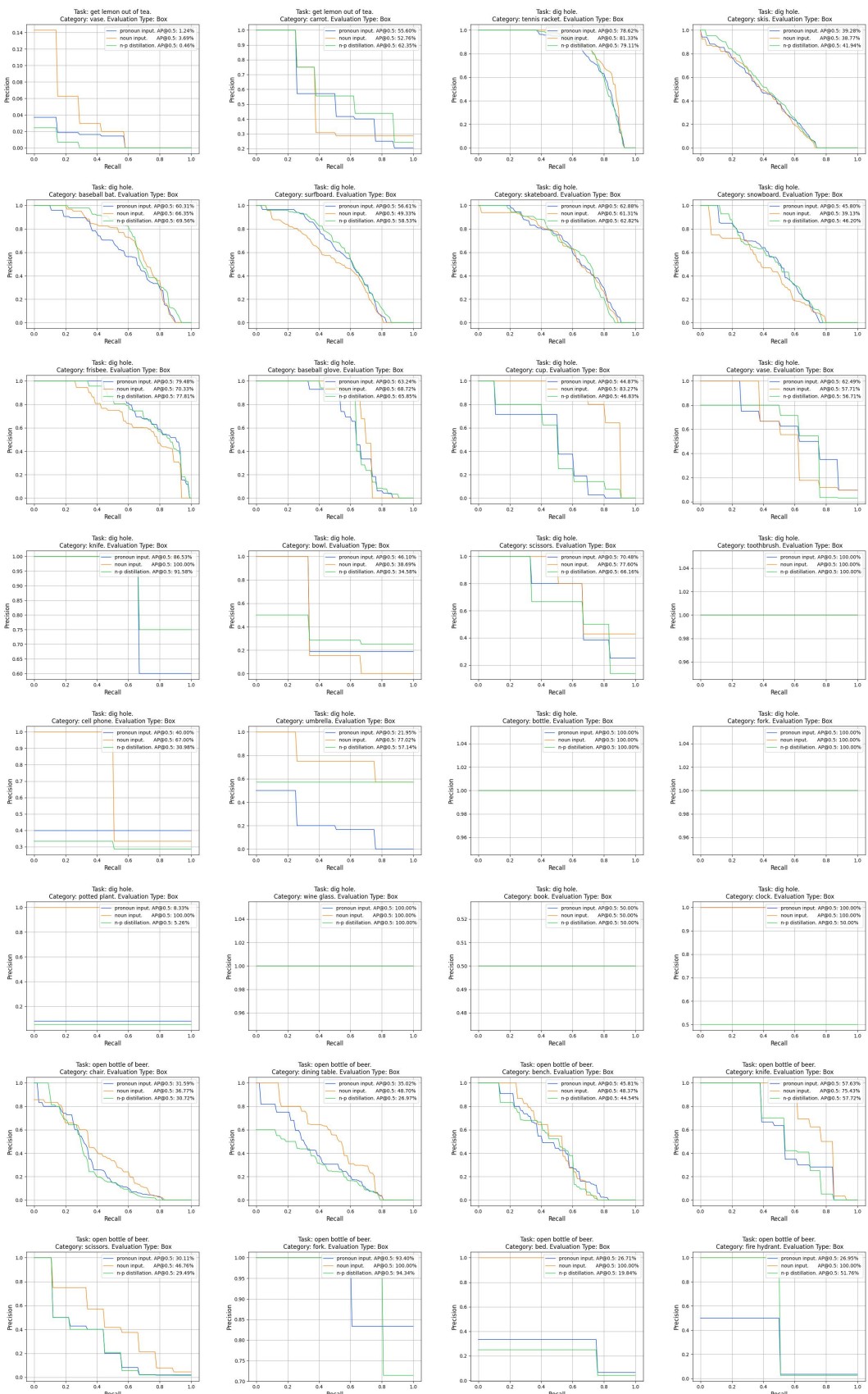

Figure 7: The precision-recall curves for object detection on the test data that contain objects of specified classes in each task (cont.).

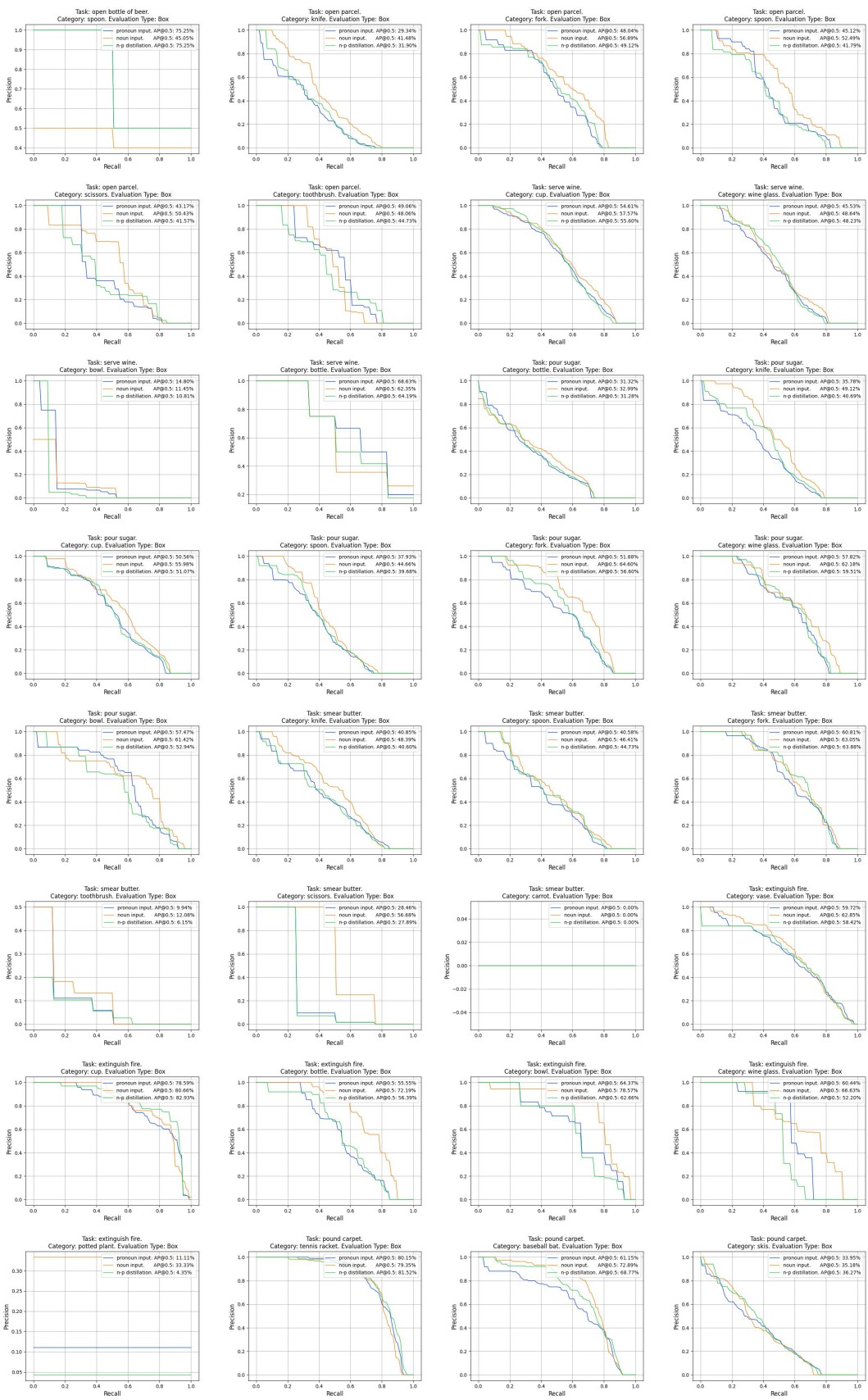

Figure 7: The precision-recall curves for object detection on the test data that contain objects of specified classes in each task (cont.).

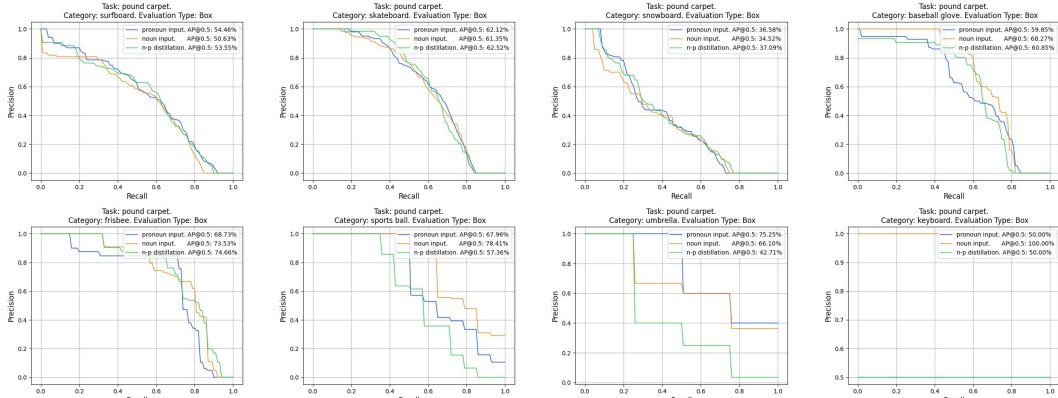

Figure 7: The precision-recall curves for object detection on the test data that contain objects of specified classes in each task (cont.).

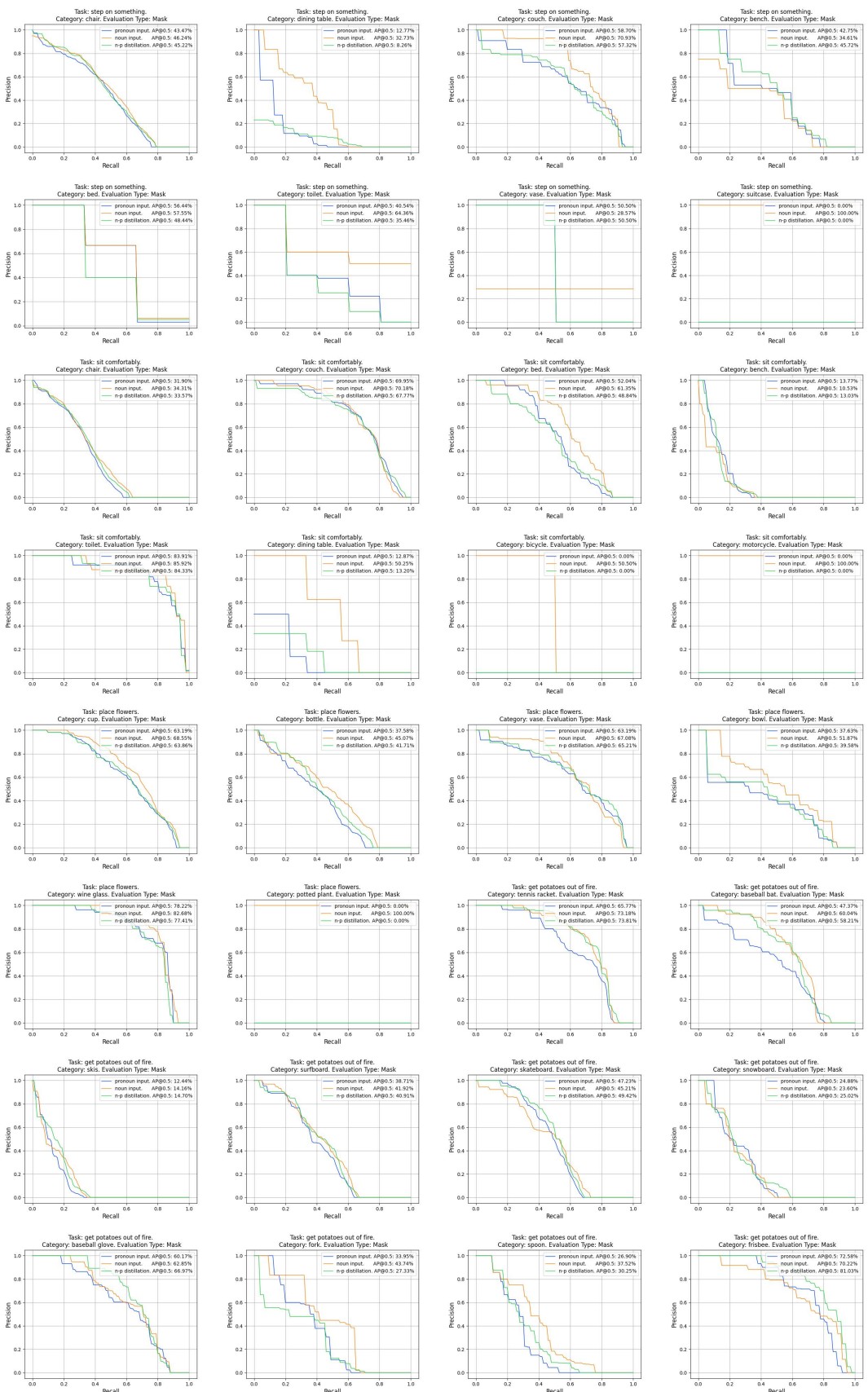

Figure 8: The precision-recall curves for instance segmentation on the test data that contain objects of specified classes in each task.

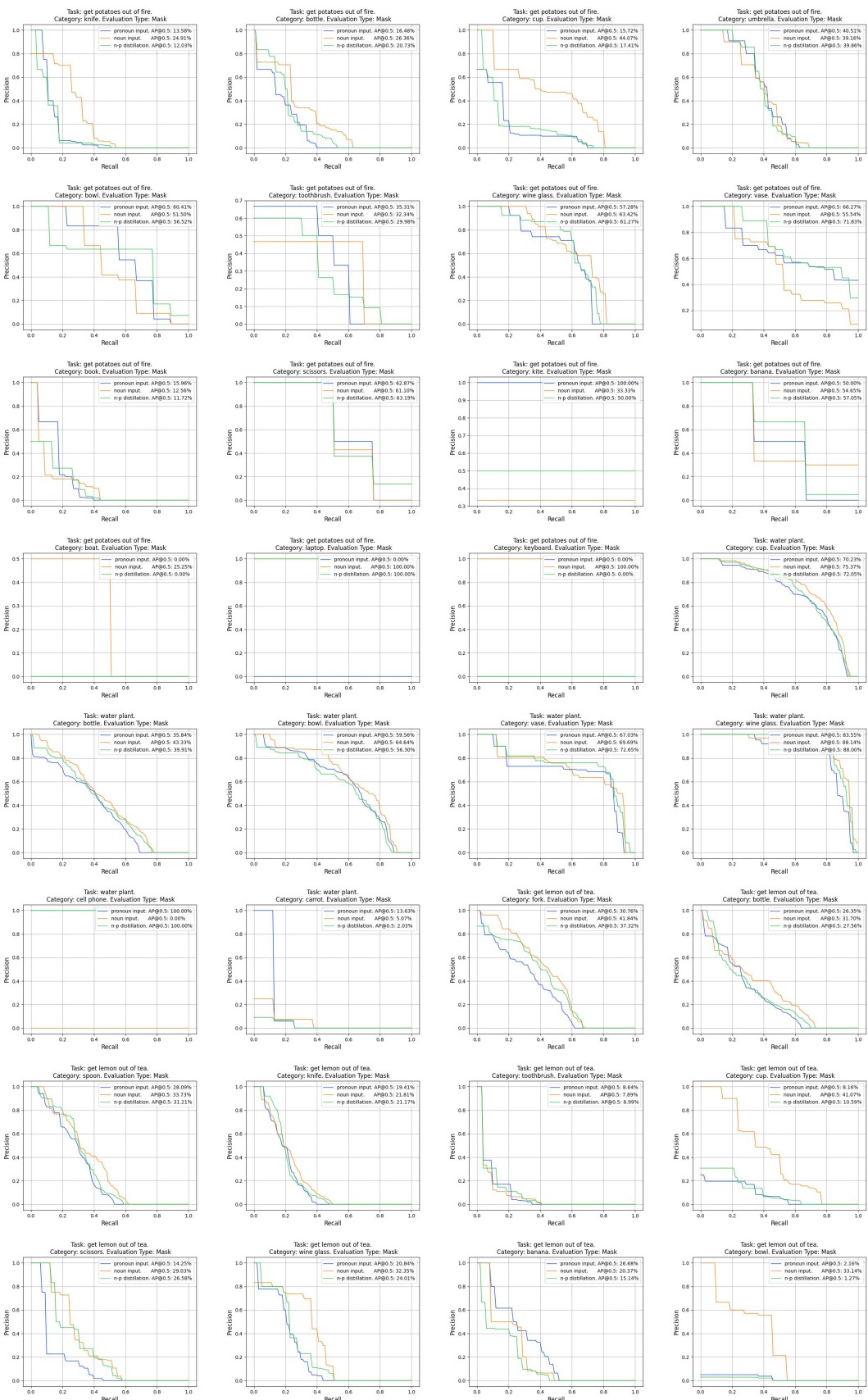

Figure 8: The precision-recall curves for instance segmentation on the test data that contain objects of specified classes in each task (cont.).

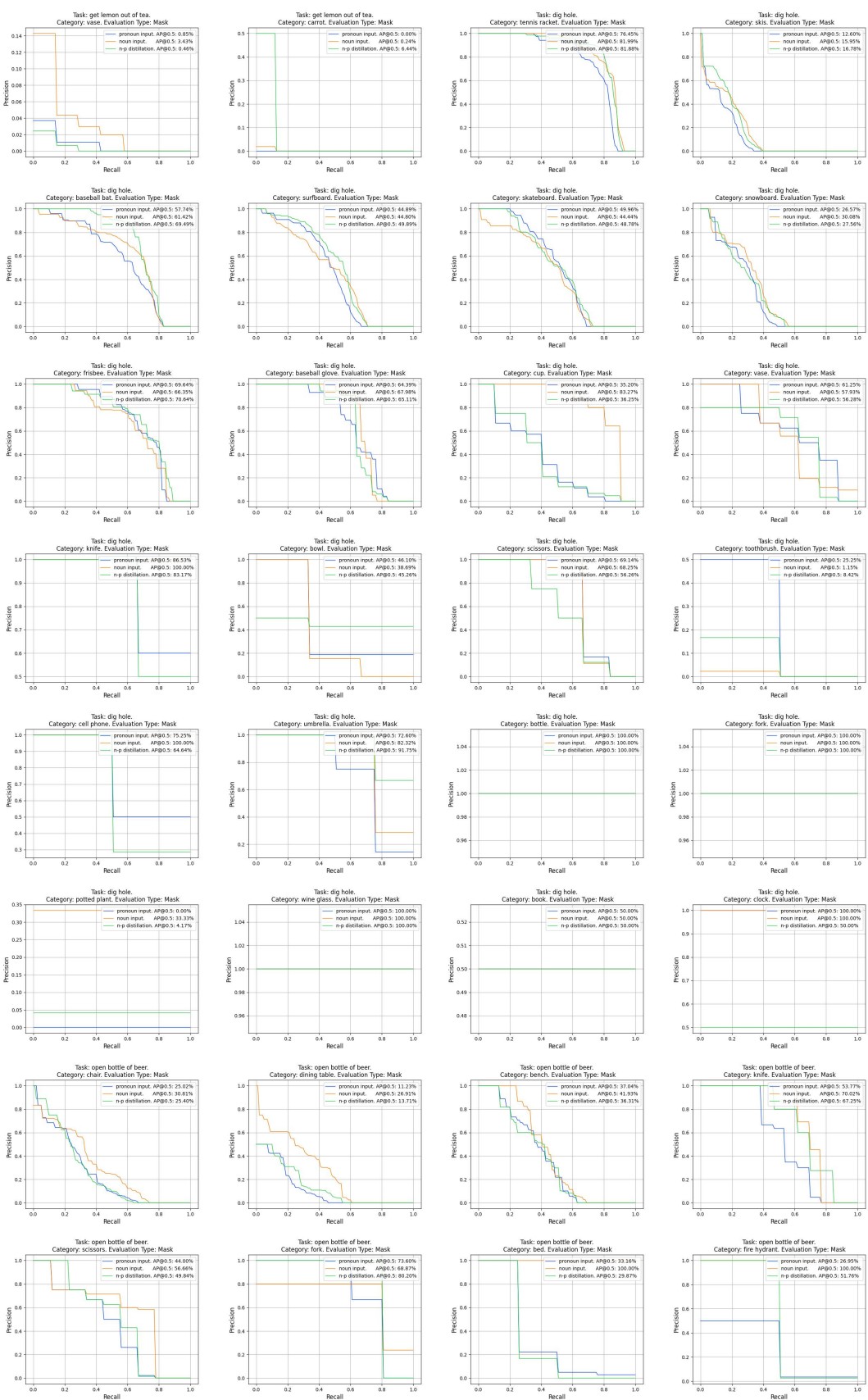

Figure 8: The precision-recall curves for instance segmentation on the test data that contain objects of specified classes in each task (cont.).

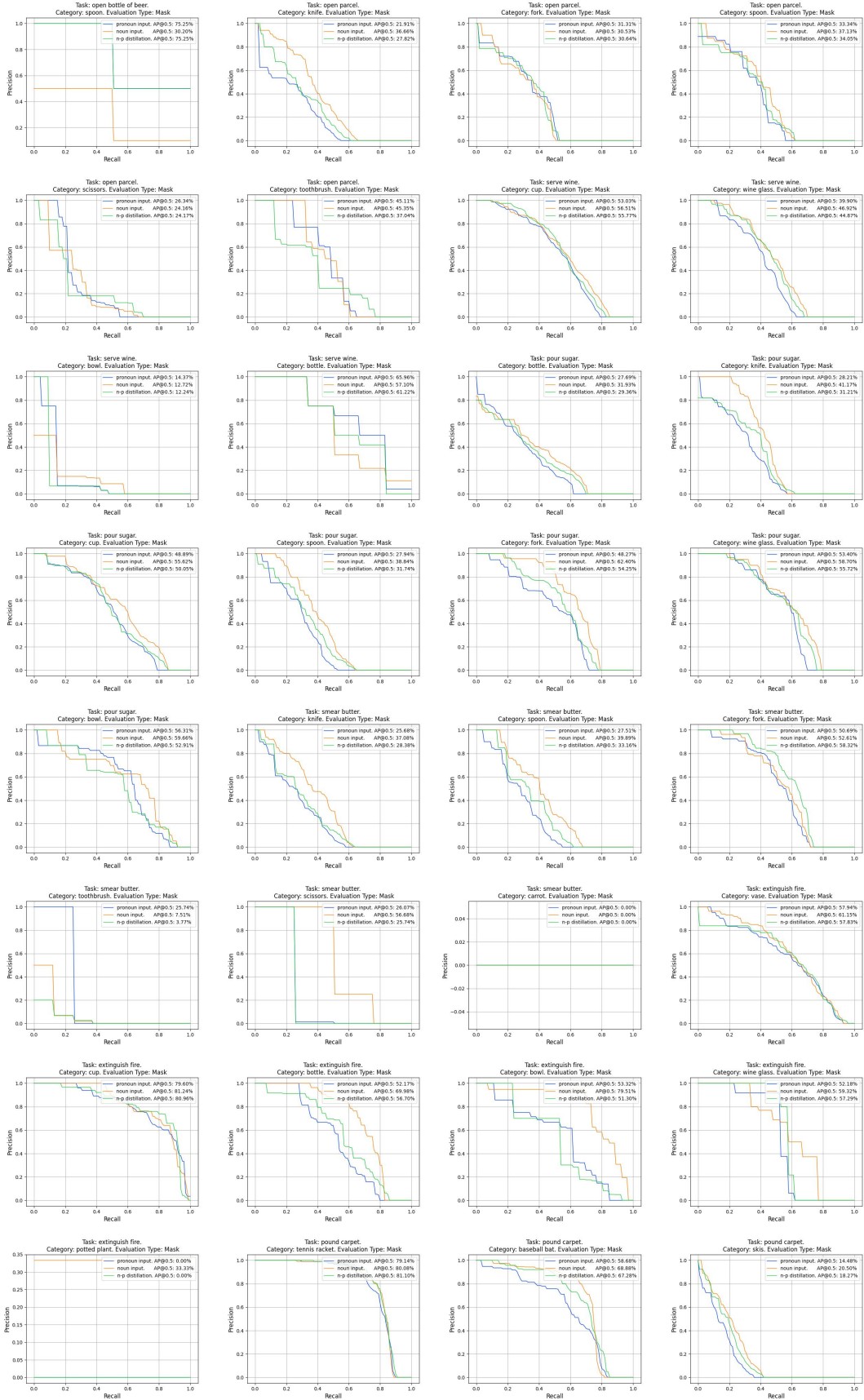

Figure 8: The precision-recall curves for instance segmentation on the test data that contain objects of specified classes in each task (cont.).

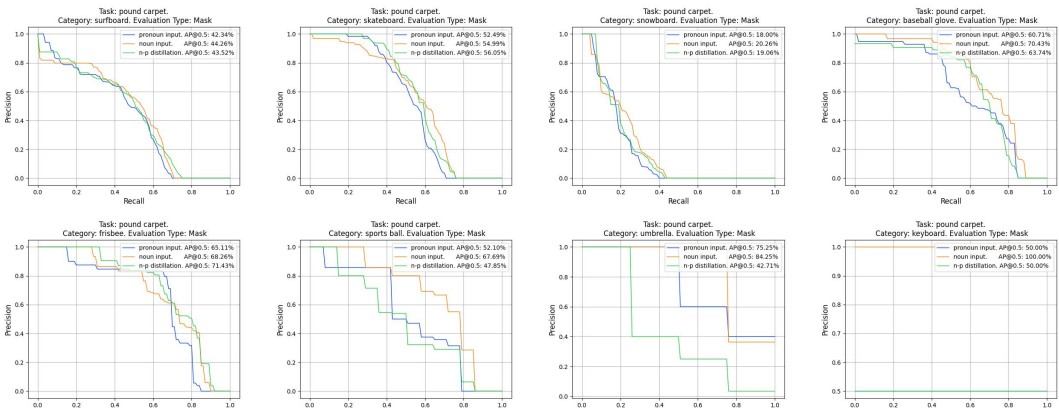

Figure 8: The precision-recall curves for instance segmentation on the test data that contain objects of specified classes in each task (cont.).

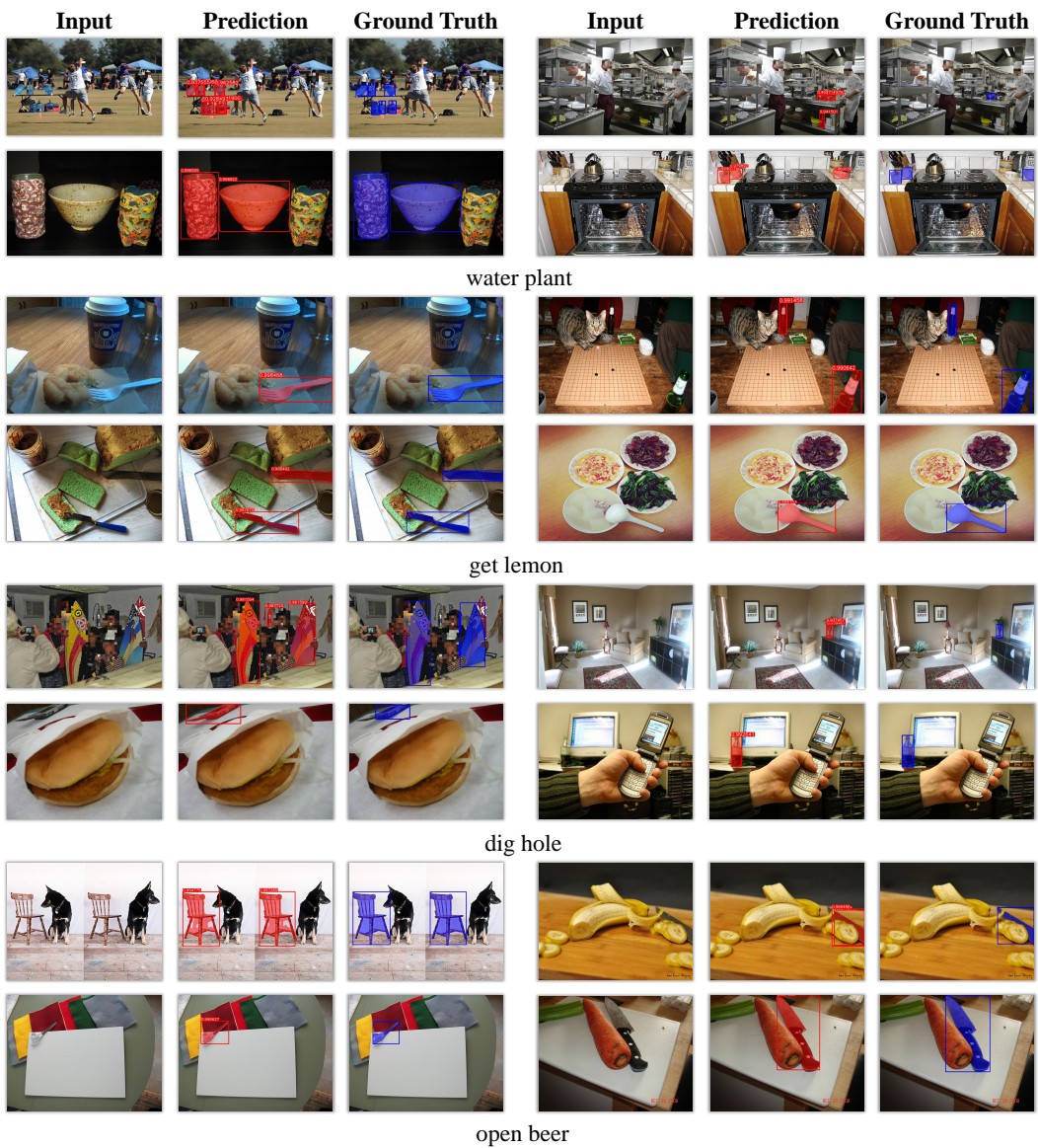

Figure 9: More qualitative results.

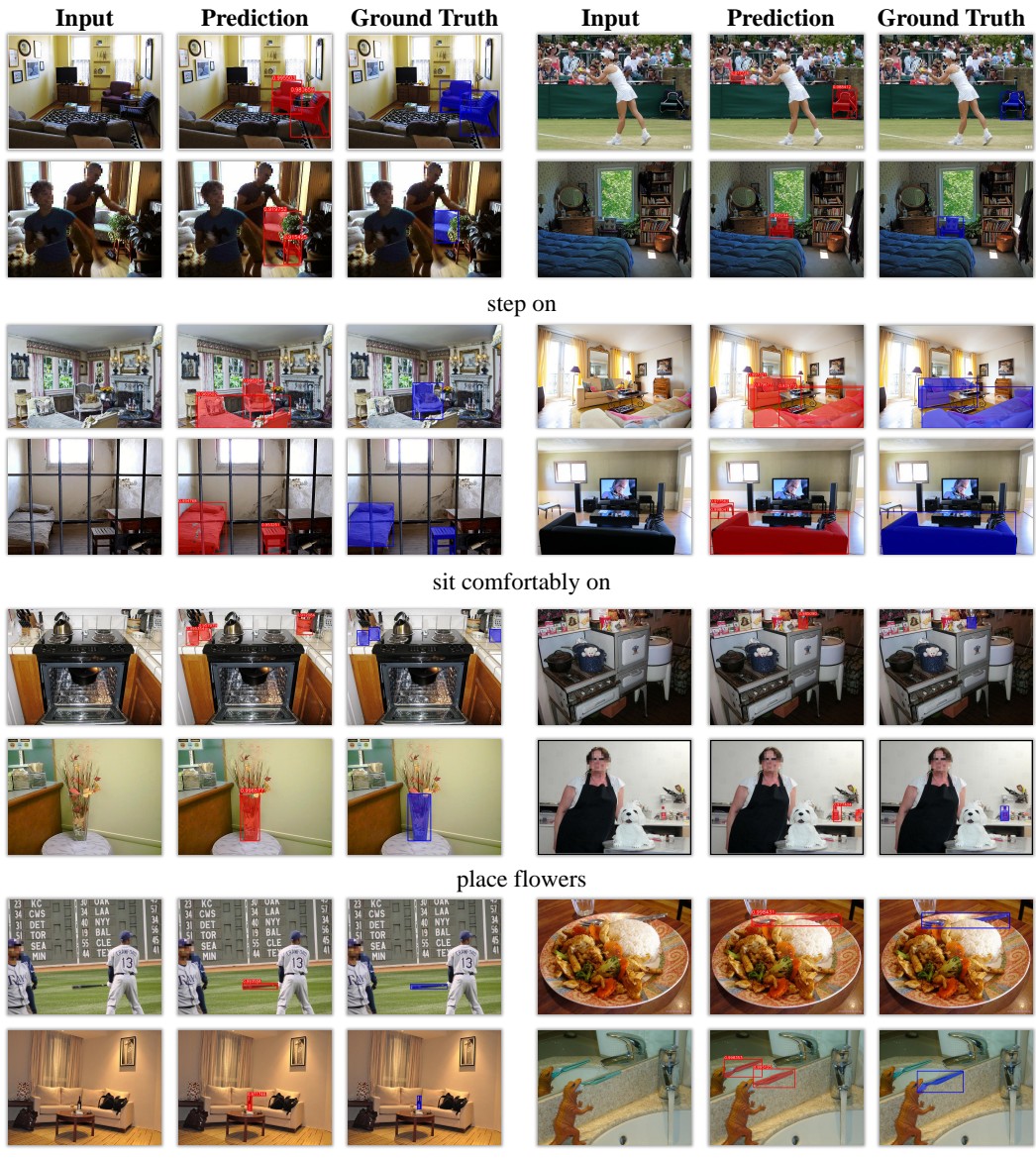

Figure 9: More qualitative results (cont.).

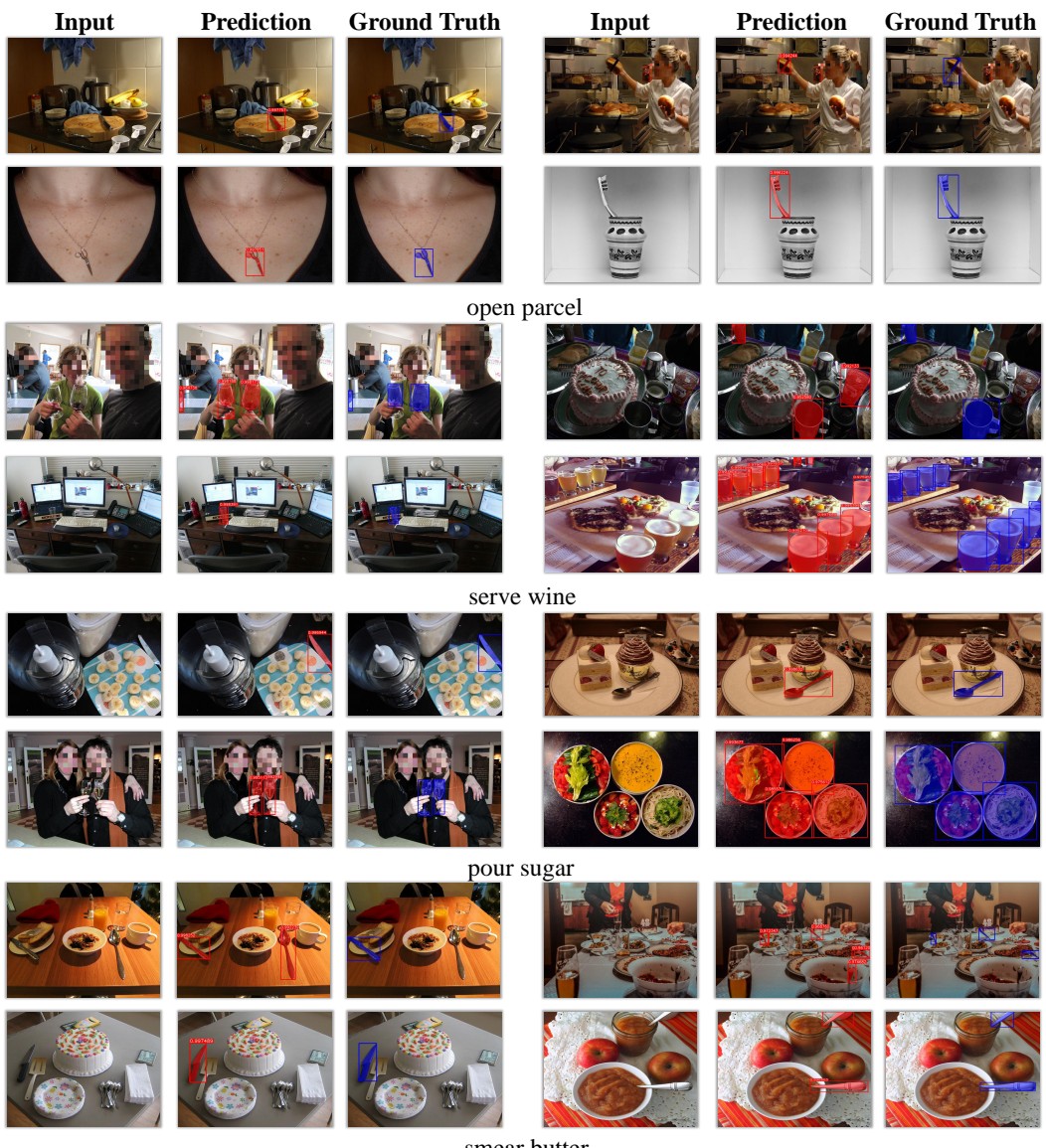

Figure 9: More qualitative results (cont.).

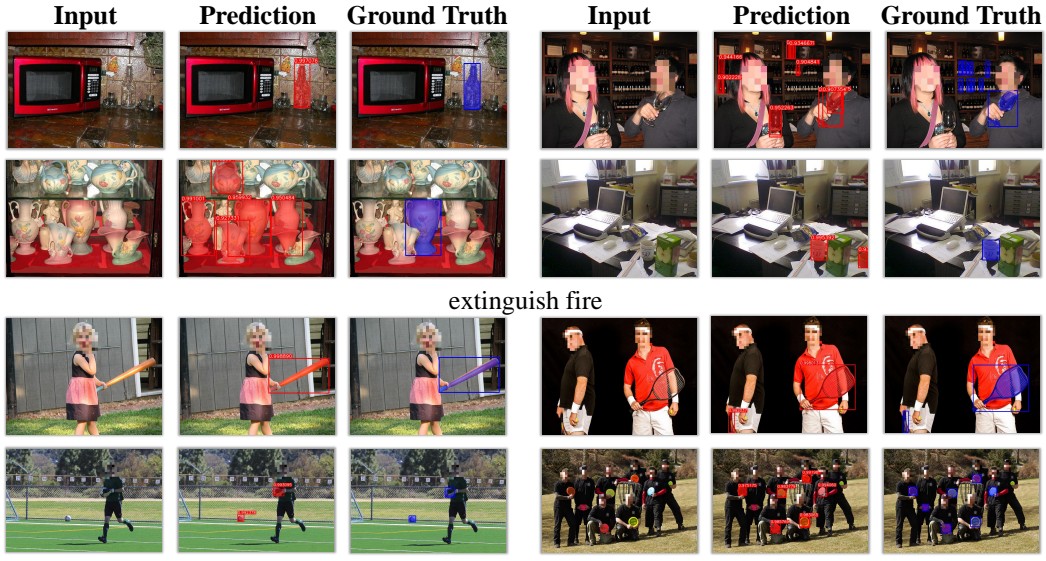

Figure 9: More qualitative results (cont.).