# OpenReview forum: "TOIST: Task Oriented Instance Segmentation Transformer with Noun-Pronoun Distillation"
_NeurIPS.cc/2022/Conference — NeurIPS 2022 Accept_

### Official Review · Reviewer_aXr4 · 2022-07-02

**Rating:** 5
**Confidence:** 4
**Soundness:** 3 good
**Presentation:** 2 fair
**Contribution:** 3 good

**Summary:**

This paper aims at task oriented detection. Instead of specifying the type of object to detect, this problem requires detection of the objects that best fits the task description. The authors proposed a largely Transformer based model, TOIST, that outperformed the previous state-of-the-art. They then proposed two distillation techniques to distill the type of object into the student model, and performance is further boosted. Experiments are performed on COCO-Tasks.

POST-REBUTTAL UPDATE:

I have read the authors' rebuttal. The authors added a lot of experiments to justify their design and showcase their big improvement over the previous baseline. However I don't think my non-result related questions are well-addressed, such as "posing questions about the baseline and the dataset in general". Overall I decided to slightly increase my score from 4 to 5. Regardless of the final result, I suggest the authors to simplify the proposed approach if possible, e.g. throwing away the notion of "task", which seems to deliver the best performance according to the new experiment.

**Questions:**

1. L169 mentioned that the text encoder is "pre-trained". What data is this text encoder pre-trained on? This is the reason why the pronoun in Table 4 makes a difference, right? What would the performance be if this part is trained from scratch? Does the distillation still work?

2. The "clustering distillation" component requires the notion of "number of tasks", and a clustering algorithm is done for each task. However, I don't think the concept of "task" is well defined in this paper (e.g. in Section 3). Does "task" equal to verb, like "dig hole" is one task, and "sit comfortably on" is another? If so, how many training examples are in COCO-Tasks, and how many tasks? Dividing the former by the latter can give the reader a rough sense of how many training examples per task, and that will also inform how the number of clusters, K, ought to be chosen.

3. Following the question above, Section 5.5 ablated the cluster number K. What about n_{task}? Does the distillation still work when n_{task} = 1, i.e. throwing away the notion of "task"?

4. I do not understand why distillation has to be done the way it is in "clustering distillation". What about loading the same data for Teacher and Student, and simply distill l_{noun} into l_{pronoun} (say in the same way as Equation 4)? This gets rid of introducing "number of tasks" and "memory bank", which greatly simplifies the proposed method.

**Limitations:**

The authors used 3 lines in Conclusion to talk about limitations. I feel it can be expanded to talk about some of the angles in my Questions section above.

**Strengths And Weaknesses:**

Strengths:
- The base model, TOIST, is fairly well motivated and well described.
- The performance gain over the previous state-of-the-art seems significant.
- The idea of distilling the object type into "something", i.e. distilling noun into pronoun, is interesting and novel in my opinion.

Weaknesses:
- The distillation version of TOIST may be a bit overly complicated, and I do not understand why it has to be designed the way it is.
- The baseline [48] is more than 3 years old, and as someone who is not extremely familiar with COCO-Tasks, it is concerning that there is no follow-up works in 3 years, posing questions about the baseline and the dataset in general.

Overall, I think originality is fairly good; quality, clarity, significance is medium.

---

> ### Author Response · Authors · 2022-08-02
> **Responses to Reviewer #aXr4 [1/3]**
>
> We thank R#aXr4 for professional feedbacks. Here we address raised concerns one by one.
>
> ### Weakness.1
> > The distillation version of TOIST may be a bit overly complicated, and I do not understand why it has to be designed the way it is.
>
> We have tried the simplified method that directly distill $l\_{\rm{pron}}^{\rm{tr}}$ into $l\_{\rm{noun}}^{\rm{tr}}$ instead of clustering distillation, but it does not work well. Please see the response to question.4 for the quantitative results.
>
> ### Weakness.2
> > The baseline [48] is more than 3 years old, and as someone who is not extremely familiar with COCO-Tasks, it is concerning that there is no follow-up works in 3 years, posing questions about the baseline and the dataset in general.
>
> In order to demonstrate that TOIST is not only stronger than methods developed 3 years ago, we present a new baseline 'MDETR+GGNN'. It takes a strong noun reference understanding model published in ICCV 2021 [1] as the detector.
>
> To leverage the knowledge in noun referring expression comprehension, we use the official pre-trained model of MDETR and then fine-tune it on the COCO-Task dataset. We use the class names of the ground truth objects in each image as the text input to detect these objects. Then we use the GGNN model [2] to infer which objects are preferred for a task. The results are shown below:
>
>
>
> Table 1: Comparison of the proposed method to 'MDETR+GGNN' baseline on the COCO-Tasks dataset.
> |              Method              | $\rm{mAP}^{box}$ | $\rm{mAP}^{mask}$ |
> |:--------------------------------:|:-----------:|:------------:|
> | MDETR+GGNN                       | 36.8        | 30.3         |
> | TOIST                            | *41.3(+4.5)*  | *35.2(+4.9)*   |
> | TOIST w/ distillation            | **43.9(+7.1)**  | **38.8(+8.5)**   |
>
> The detection and segmentation results for each class are:
>
> Table 2: The object detection results of 'MDETR+GGNN' and the proposed method for each task.
> |         Method        |   1  |   2  |   3  |   4  |   5  |   6  |   7  |   8  |   9  |  10  |  11  |  12  |  13  |  14  | mean |
> |:---------------------:|:----:|:----:|:----:|:----:|:----:|:----:|:----:|:----:|:----:|:----:|:----:|:----:|:----:|:----:|:----:|
> | MDETR+GGNN            | *44.3* | 36.5 | 45.2 | 28.6 | 44.0 | **27.6** | 35.9 | 20.7 | **34.7** | *46.3* | 27.8 | 41.5 | 46.5 | 36.2 | 36.8 |
> | TOIST                 | 44.0 | *39.5* | *46.7* | *43.1* | *53.6* | 23.5 | *52.8* | *21.3* | 32.0 | *46.3* | *33.1* | *41.7* | *48.1* | *52.9* | *41.3* |
> | TOIST w/ distillation | **46.2** | **39.6** | **49.9** | **47.1** | **54.5** | *26.7* | **57.3** | **23.1** | *33.1* | **49.9** | **35.4** | **44.7** | **52.1** | **54.9** | **43.9** |
>
> Table 3: The instance segmentation results of 'MDETR+GGNN' and the proposed method for each task.
> |         Method        |   1  |   2  |   3  |   4  |   5  |   6  |   7  |   8  |   9  |  10  |  11  |  12  |  13  |  14  | mean |
> |:---------------------:|:----:|:----:|:----:|:----:|:----:|:----:|:----:|:----:|:----:|:----:|:----:|:----:|:----:|:----:|:----:|
> | MDETR+GGNN            | 36.9 | 31.3 | 43.6 | 17.1 | 42.9 | *20.1* | 19.9 | *18.7* | *24.5* | *45.5* | 23.1 | *30.9* | 46.2 | 24.0 | 30.3 |
> | TOIST                 | *37.0* | *34.4* | *44.7* | *34.2* | *51.3* | 18.6 | *40.5* | 17.1 | 23.4 | 43.8 | *29.3* | 29.9 | *46.6* | *42.4* | *35.2* |
> | TOIST w/ distillation | **40.8** | **36.5** | **48.9** | **37.8** | **53.4** | **22.1** | **44.4** | **20.3** | **26.9** | **48.1** | **31.8** | **34.8** | **51.5** | **46.3** | **38.8** |
>
> Note that this baseline is also tested with privileged noun ground truth, but our distillation method only use the priviledged knowledge during training. Nevertheless, our proposed method has a significant performance improvement over this strong baseline.
>
> We will add the results into the supplementary.
>
> As for the dataset, we have included some statistics in the Figure 1-8 in the supplementary (we mark the corresponding analysis in the supplementary in red). These statistics demonstrate the diversity and complexity of the COCO-Tasks dataset.

---

> ### Author Response · Authors · 2022-08-02
> **Responses to Reviewer #aXr4 [2/3]**
>
>
> ### Question.1
> > L169 mentioned that the text encoder is "pre-trained". What data is this text encoder pre-trained on? This is the reason why the pronoun in Table 4 makes a difference, right? What would the performance be if this part is trained from scratch? Does the distillation still work?
>
> The text encoder is a RoBERTa-base [3], which is pre-trained on five datasets: BookCorpus, English Wikipedia, CC-News, OpenWebText, and Stories. The implementation and weights are taken from HuggingFace [4].
>
> To verify the effectiveness of our distillation, we have trained our model from scratch on the COCO-Tasks dataset.
> The final results are:
>
>
>
> Table 4: The results of TOIST without pre-training.
> |              Method              | $\rm{mAP}^{box}$ | $\rm{mAP}^{mask}$ |
> |:--------------------------------:|:-----------:|:------------:|
> | pronoun input                    | 3.65        | 5.74            |
> | noun input                       | 11.19       | 12.67   |
> | noun-pronoun distillation                 | 7.43(+3.78)        | 11.28(+5.54)   |
>
>
>
> The results without pre-training demonstrates that the proposed distillation can still work well even without a pre-trained text encoder.
>
> We will add the results into the supplementary.
>
> ### Question.2
> > The "clustering distillation" component requires the notion of "number of tasks", and a clustering algorithm is done for each task. However, I don't think the concept of "task" is well defined in this paper (e.g. in Section 3). Does "task" equal to verb, like "dig hole" is one task, and "sit comfortably on" is another? If so, how many training examples are in COCO-Tasks, and how many tasks? Dividing the former by the latter can give the reader a rough sense of how many training examples per task, and that will also inform how the number of clusters, K, ought to be chosen.
>
> Yes, in this paper, every 'task' corresponds to a verb phrase like 'dig hole' or 'sit comfortably on'.
> As mentioned in line 263-264, there are 14 tasks contained in the COCO-Tasks dataset, and for each task, there are 3600 train images and 900 test images.
> More details about the dataset can be found in section 2 of the supplementary.
> Furthermore, we have included some statistics about the categories of the ground truth objects in each task in the Figure 1-2 in the supplementary, which show the diversity of the data distribution.
> We also present exhaustive class-by-class quantitative results of our proposed method in Figure 3-8 in the supplementary. The corresponding analysis is marked in red, which demonstrates the effectiveness of our method
>
>
> ### Question.3
> > Following the question above, Section 5.5 ablated the cluster number K. What about $n\_{\rm{task}}$? Does the distillation still work when $n\_{\rm{task}}$ = 1, i.e. throwing away the notion of "task"?
>
> We thank R#aXr4 for this suggestion. We have included a new ablation study about the $n\_{\rm{task}}$. We show the object detection results on 5 tasks below:
>
> Table 5: Ablations for the task number $n\_{\rm{task}}$ on object detection.
> |          Method          | step on something | sit comfortably | place flowers | get potatoes out of fire | water plant |
> |:------------------------:|:-----------------:|:---------------:|:-------------:|:------------------------:|:-----------:|
> |       TOIST w/o dis      |        44.0       |       39.5      |      46.7     |           43.1           |     53.6    |
> | dis $n\_{\rm{task}}$ = 14 |     46.2(+2.2)    |    39.6(+0.1)   |   49.9(+3.2)  |        *47.1(+4.0)*        |  54.5(+0.9) |
> |  dis $n\_{\rm{task}}$ = 5 |     *46.4(+2.4)*    |    *40.7(+1.2)*   |   **51.3(+4.6)**  |        46.8(+3.7)        |  *54.6(+1.0)* |
> |  dis $n\_{\rm{task}}$ = 1 |     **47.0(+3.0)**    |    **42.1(+2.6)**   |   *50.8(+4.1)*  |            **47.4(+4.3)**           |     **55.2(+1.6)**     |
>
> In this table, the first line corresponds to the plain TOIST without distillation, and the other lines show the results of distillation with different $n\_{\rm{task}}$.
> The results demonstrate that our proposed distillation still works for different $n\_{\rm{task}}$, even if $n\_{\rm{task}}$ = 1.
> And overall, smaller $n\_{\rm{task}}$ leads to better performance. We attribute this to the reduced problem complexity due to the less interaction between different tasks, which makes it easier to improve the ability of the model to understand verbs through noun-pronoun distillation.
>
> We will add the results into the supplementary.

---

> ### Author Response · Authors · 2022-08-02
> **Responses to Reviewer #aXr4 [3/3]**
>
>
> ### Question.4
> > I do not understand why distillation has to be done the way it is in "clustering distillation". What about loading the same data for Teacher and Student, and simply distill l\_{noun} into l\_{pronoun} (say in the same way as Equation 4)? This gets rid of introducing "number of tasks" and "memory bank", which greatly simplifies the proposed method.
>
> We have tried this simplified method that directly distill $l\_{\rm{pron}}^{\rm{tr}}$ into $l\_{\rm{noun}}^{\rm{tr}}$, but it does not work well. The quantitative results are demonstrated below:
>
> Table 6: Comparison of different distillation methods.
>
> |              Method              | $\rm{mAP}^{box}$ | $\rm{mAP}^{mask}$ |
> |:--------------------------------:|:-----------:|:------------:|
> | TOIST                            | 41.3        | 35.2         |
> | distill from $l\_{c\_s}^j$ to $l\_{\rm{pron}}^{\rm{tr}}$            | **43.9(+2.6)**  | **38.8(+3.6)**   |
> | distill from $l\_{\rm{noun}}^{\rm{tr}}$ to $l\_{\rm{pron}}^{\rm{tr}}$ | *41.9(+0.6)*  | *36.0(+0.8)*   |
>
> Therefore, we propose the clustering distillation method. And the overhead introduced by this method is only for maintaining the memory bank, doing the cluster selection and calculating the cluster loss. The process is clear and the extra cost is significantly smaller than the model backbone, while improving the performance markedly.
>
> We will present the result in the new version of the paper.
>
>
> ### References
> [1] Kamath A, Singh M, LeCun Y, et al. MDETR-modulated detection for end-to-end multi-modal understanding[C]//Proceedings of the IEEE/CVF International Conference on Computer Vision. 2021: 1780-1790.
>
> [2] Sawatzky J, Souri Y, Grund C, et al. What object should i use?-task driven object detection[C]//Proceedings of the IEEE/CVF Conference on Computer Vision and Pattern Recognition. 2019: 7605-7614.
>
> [3] Liu Y, Ott M, Goyal N, et al. Roberta: A robustly optimized bert pretraining approach[J]. arXiv preprint arXiv:1907.11692, 2019.
>
> [4] Wolf T, Debut L, Sanh V, et al. Transformers: State-of-the-art natural language processing[C]//Proceedings of the 2020 conference on empirical methods in natural language processing: system demonstrations. 2020: 38-45.

---

> ### Author Response · Authors · 2022-08-08
> **Response to Reviewer #aXr4**
>
> Dear reviewer,
>
> Please let us know if our responses have addressed the issues raised in your review. We hope that our corrections, clarifications, and additional results address the concerns you've raised. We are happy to address any further concerns.

---

### Official Review · Reviewer_mBEm · 2022-07-12

**Rating:** 5
**Confidence:** 3
**Soundness:** 3 good
**Presentation:** 3 good
**Contribution:** 2 fair

**Summary:**

In order to handle the affordance recognition task, this paper proposes a Task-Oriented Instance Segmentation Transformer (TOIST) to find objects that best afford an action indicated by verbs. The TOIST is a teacher-student knowledge distillation model, and such a model leverages the referring expression comprehension algorithm as the teacher module for guiding the student module to learn the noun-pronoun transformation. The experiments show the positive effect of the knowledge distillation mechanism.

**Questions:**

The tackled affordance recognition task is not a well-explored research topic; hence, the compared baseline method [48] is not advanced. In order to demonstrate the performance gain of the proposed TOIST, it is better to train the model without using the extra training data and comparing it with the baseline of an advanced backbone network, for example, ‘mdetr+GGNN.’ Please see [Weaknesses] for reference.

**Limitations:**

The authors described the limitations and potential negative societal impact of their work.

**Strengths And Weaknesses:**

[Strengths]
+ The idea of utilizing the referring expression comprehension algorithm as the teacher module is interesting.
+ The manuscript is well organized and has several interesting analyses.

[Weaknesses]
- The contribution to upgrading the task-oriented detection into task-oriented instance segmentation upon the `existing’ transformer model is weak.
- Though the existing method [48] is a two-stage model, yet the proposed TOIST needs to separately fine-turn the pre-trained student and teacher TOIST models with a final knowledge distilling. It seems that the pre-trained models employ the extra data for training, and hence the extra training data and the extra training procedure make the advantage of the claimed one-stage model somewhat weak.
- Compared with the state-of-the-art methods, it is not fair to compare the methods extracting features with different backbones. It is interesting whether the ‘TOIST w/ distillation’ in table 2 can still surpass the baseline with the same backbone, i.e., ‘mdetr+GGNN’?

---

> ### Author Response · Authors · 2022-08-02
> **Responses to Reviewer #mBEm [1/2]**
>
> We thank R#mBEm for professional feedbacks. Here we address raised concerns one by one.
>
> ### Weakness.1
> > The contribution to upgrading the task-oriented detection into task-oriented instance segmentation upon the 'existing' transformer model is weak.
>
> We would like to respond to this contribution concern from three perspectives:
>
> (1) In order to investigate whether our noun-pronoun distillation training framework is a standalone technical contribution without pre-trained models, we present experiments without using pre-trained models. The quantitative results can be found in the Weakness.2 part.
>
> (2) In order to investigate whether our TOIST architecture is a standalone technical contribution by marginalizing the benefits brought by pre-trained models, we present another baseline 'MDETR+GGNN'. The quantitative results can be found in the Weakness.3 part.
>
> (3) Finally, except for technical contributions, we think this study has a methodological contribution: we propose the new scheme of using verb-pronoun for task oriented detection, and using pre-trained transformer models is its implementation. We reformulate the problem into a verb reference understanding one so that the noun-pronoun distillation becomes possible.
>
> ### Weakness.2
> > Though the existing method [48] is a two-stage model, yet the proposed TOIST needs to separately fine-turn the pre-trained student and teacher TOIST models with a final knowledge distilling. It seems that the pre-trained models employ the extra data for training, and hence the extra training data and the extra training procedure make the advantage of the claimed one-stage model somewhat weak.
>
> To verify the effectiveness of our distillation, we have trained our model from scratch on the COCO-Tasks dataset.
> The final results are:
>
> Table 1: The results of TOIST without pre-training.
> |              Method              | $\rm{mAP}^{box}$ | $\rm{mAP}^{mask}$ |
> |:--------------------------------:|:-----------:|:------------:|
> | pronoun input                    | 3.65        | 5.74            |
> | noun input                       | 11.19       | 12.67   |
> | noun-pronoun distillation                 | 7.43(+3.78)        | 11.28(+5.54)   |
>
> The results demonstrates that the proposed distillation can still work well even without pre-training.
>
> We will add the results into the supplementary.

---

> ### Author Response · Authors · 2022-08-02
> **Responses to Reviewer #mBEm [2/2]**
>
>
>
> ### Weakness.3
> > Compared with the state-of-the-art methods, it is not fair to compare the methods extracting features with different backbones. It is interesting whether the 'TOIST w/ distillation' in table 2 can still surpass the baseline with the same backbone, i.e., 'mdetr+GGNN'?
>
> We thank R#mBEm for this suggestion. We have included the new baseline 'MDETR+GGNN'.
>
> To leverage the knowledge in noun referring expression comprehension, we use the official pre-trained model of MDETR and then fine-tune it on the COCO-Task dataset. We use the class names of the ground truth objects in each image as the text input to detect these objects. Then we use the GGNN model [2] to infer which objects are preferred for a task. The results are shown below:
>
> Table 2: Comparison of the proposed method to 'MDETR+GGNN' baseline on the COCO-Tasks dataset.
> |              Method              | $\rm{mAP}^{box}$ | $\rm{mAP}^{mask}$ |
> |:--------------------------------:|:-----------:|:------------:|
> | MDETR+GGNN                       | 36.8        | 30.3         |
> | TOIST                            | *41.3(+4.5)*  | *35.2(+4.9)*   |
> | TOIST w/ distillation            | **43.9(+7.1)**  | **38.8(+8.5)**   |
>
> The detection and segmentation results for each class are:
>
> Table 3: The object detection results of 'MDETR+GGNN' and the proposed method for each task.
> |         Method        |   1  |   2  |   3  |   4  |   5  |   6  |   7  |   8  |   9  |  10  |  11  |  12  |  13  |  14  | mean |
> |:---------------------:|:----:|:----:|:----:|:----:|:----:|:----:|:----:|:----:|:----:|:----:|:----:|:----:|:----:|:----:|:----:|
> | MDETR+GGNN            | *44.3* | 36.5 | 45.2 | 28.6 | 44.0 | **27.6** | 35.9 | 20.7 | **34.7** | *46.3* | 27.8 | 41.5 | 46.5 | 36.2 | 36.8 |
> | TOIST                 | 44.0 | *39.5* | *46.7* | *43.1* | *53.6* | 23.5 | *52.8* | *21.3* | 32.0 | *46.3* | *33.1* | *41.7* | *48.1* | *52.9* | *41.3* |
> | TOIST w/ distillation | **46.2** | **39.6** | **49.9** | **47.1** | **54.5** | *26.7* | **57.3** | **23.1** | *33.1* | **49.9** | **35.4** | **44.7** | **52.1** | **54.9** | **43.9** |
>
> Table 4: The instance segmentation results of 'MDETR+GGNN' and the proposed method for each task.
> |         Method        |   1  |   2  |   3  |   4  |   5  |   6  |   7  |   8  |   9  |  10  |  11  |  12  |  13  |  14  | mean |
> |:---------------------:|:----:|:----:|:----:|:----:|:----:|:----:|:----:|:----:|:----:|:----:|:----:|:----:|:----:|:----:|:----:|
> | MDETR+GGNN            | 36.9 | 31.3 | 43.6 | 17.1 | 42.9 | *20.1* | 19.9 | *18.7* | *24.5* | *45.5* | 23.1 | *30.9* | 46.2 | 24.0 | 30.3 |
> | TOIST                 | *37.0* | *34.4* | *44.7* | *34.2* | *51.3* | 18.6 | *40.5* | 17.1 | 23.4 | 43.8 | *29.3* | 29.9 | *46.6* | *42.4* | *35.2* |
> | TOIST w/ distillation | **40.8** | **36.5** | **48.9** | **37.8** | **53.4** | **22.1** | **44.4** | **20.3** | **26.9** | **48.1** | **31.8** | **34.8** | **51.5** | **46.3** | **38.8** |
>
> Note that this baseline is also tested with privileged noun ground truth, but our distillation method only use the privileged knowledge during training. Nevertheless, our proposed method still has a significant performance improvement over this strong baseline.
>
> We will add the results into the supplementary.
>
> ### Question.1
> > The tackled affordance recognition task is not a well-explored research topic; hence, the compared baseline method [48] is not advanced. In order to demonstrate the performance gain of the proposed TOIST, it is better to train the model without using the extra training data and comparing it with the baseline of an advanced backbone network, for example, 'mdetr+GGNN.' Please see [Weaknesses] for reference.
>
> Please see the responses to weakness.2 and weakness.3.

---

> > ### Comment · Reviewer_mBEm · 2022-08-08
> > **Thanks for the authors' response.**
> >
> > Thanks for the authors' response.
> > * As shown in Table 1 vs. Table 2, it is known that the TOIST requires the pre-trained models, trained with the extra data, to boost the model performance.
> > * In Table 2, do TOIST and MDETR+GGNN employ the same image encoder and text encoder? Could the performance also derive from the extra training data used in the pre-trained student and teacher TOIST models?

---

> > > ### Author Response · Authors · 2022-08-09
> > > **Responses to Reviewer #mBEm**
> > >
> > > Thanks for R#mBEm's feedback.
> > >
> > > Firstly, TOIST and MDETR+GGNN employ the same image encoder and text encoder. And the performance of MDETR+GGNN could derive from the pre-training process. The experimental results are shown below.
> > >
> > > Table 5: Comparison of different methods with the same backbone.
> > > |              Method              | $\rm{mAP}^{box}$ | $\rm{mAP}^{mask}$ |
> > > |:--------------------------------:|:-----------:|:------------:|
> > > | MDETR+GGNN w/o pretraining       | 9.6        | 8.6         |
> > > | MDETR+GGNN                       | 36.8        | 30.3         |
> > > | TOIST                            | *41.3*  | *35.2*   |
> > > | TOIST w/ distillation            | **43.9**  | **38.8**   |
> > >
> > > The results also show that though MDETR+GGNN can also benefit from the pretraining process, our method (with the same pretraining) still outperforms it by +7.1% $\rm{mAP}^{box}$ and +8.5% $\rm{mAP}^{mask}$. This demonstrates that our TOIST architecture is a standalone technical contribution towards task oriented instance segmentation and pretraining is necessary but insufficient to get the performance level of TOIST.
> > >
> > > Secondly, the results in Table.1 and Table.2 show that our noun-pronoun distillation training framework is a standalone technical contribution no matter whether the pretraining is used. This method makes it possible to leverage the abundant information in the well-studied noun referring expression understanding to advance the research on verb referring expression understanding.
> > >
> > > Finally, pretraining is a widely used process to improve performance on downstream tasks. We think the technical contributions and the methodological contributions of our proposed method should not be ignored just because the existence of pretraining, especially when the quantitative results (Table.5) have shown that our method still outperforms other methods in a fair comparison (with the same backbone and the same pretraining).

---

> ### Author Response · Authors · 2022-08-08
> **Response to Reviewer #mBEm**
>
> Dear reviewer,
>
> Please let us know if our responses have addressed the issues raised in your review. We hope that our corrections, clarifications, and additional results address the concerns you've raised. We are happy to address any further concerns.

---

### Official Review · Reviewer_u6fT · 2022-07-25

**Rating:** 7
**Confidence:** 3
**Soundness:** 3 good
**Presentation:** 3 good
**Contribution:** 3 good

**Summary:**

Authors propose a novel way to do task oriented object detection. Dataset used is COCO Task. Author modify the backbone used in DETR model to feed the transformer encoder with textual features along with image features so that better contextualized representations are obtained. The loss function is optimized to perform accurate bounding box localization along with instance segmentation. The paper leverages knowledge from a model trained with verb-noun captions (using the ground truth nouns) to train a student model with verb-pronoun caption. This way the model still remains noun-agnostic during inference time. But it can detect the noun from verb-pronoun if it was trained properly.

**Questions:**

1. How many unique pronouns are used in the captions to train the student model? How many objects in Coco Task dataset? Is the coco task dataset captions modified in any way (like replacing objects with pronouns) to train the student model?
2. It would be good to see an analysis of what verbs are associated with what objects (comparing ground truth and model predictions). Something like a distribution plot that the verb "sit on" was associated with "chair" in 10 out of 20 times, it was associated with "table" in 5 out of 20 times etc. That will indicate if the model fails for any verbs more frequently than others.
3. How is the score s_i is used in loss function in equation 3? Is it used in localization loss terms or segmentation loss terms? Or is it used in some other way?
4. How is default value of n_max = 256 decided? What is the value of npred? I am assuming it should be greater than the total number of objects in CoCo dataset. Is that the case?
5. Ablations for including vs not including the loss terms L_token and L_align?
6. It is not clear what G_npred means in line 245. Please define what it stands for.
7. L_match was a loss term in DETR model to encourage matching the class ad bounding boxes of ground truth and prediction.
However it is not included in loss functions here (equations 3 and 9). Why is L_match not used in loss function? In line 247, authors mention KL divergence is also a part of L_match. However, the original DETR paper doesnt mention that.
8. Instead of minimizing the distance between l_pron_tr and l_cs_j in equation 4, why can't one minimize the distance between l_pron_tr and l_noun_tr for knowledge distillation?

**Limitations:**

Limitations are briefly discussed in "Conclusion" section.

**Strengths And Weaknesses:**

 Strength:
 1. Authors propose a novel way for task oriented detection by introducing verb pronoun captions and leveraging knowledge distillation to learn from noun ground truth.
 2. The paper presents state of art results on the said task.
 3. Ablation provided to show utility of the distillation components which is one of the novelties of the paper.

Weakness:
1. The memory bank is a queue and is updated in a FIFO fashion. However, this might lead to removal of a noun feature not adequately represented in the rest of the list. Wouldn't it make more sense to update the queue by removing elements in a smarter way to reduce occurrence of similar features? For example, for any new object feature remove one of the past object features whose representation is closest to it.
2. Some ablation studies regarding why some loss terms are useful are missing.

---

> ### Author Response · Authors · 2022-08-02
> **Responses to Reviewer #u6fT [1/3]**
>
> We thank R#u6fT for professional feedbacks. Here we address raised concerns one by one.
>
>
> ### Weakness.1
> > The memory bank is a queue and is updated in a FIFO fashion. However, this might lead to removal of a noun feature not adequately represented in the rest of the list. Wouldn't it make more sense to update the queue by removing elements in a smarter way to reduce occurrence of similar features? For example, for any new object feature remove one of the past object features whose representation is closest to it.
>
> We thank R#u6fT for this suggestion. We have included a new ablation study that demonstrates the impact of this new memory updating scheme, as shown below:
>
> Table 1: Comparison of different updating methods for the memory bank in the distillation.
>
> |              Method              | $\rm{mAP}^{box}$ | $\rm{mAP}^{mask}$ |
> |:--------------------------------:|:-----------:|:------------:|
> | TOIST w/o distillation           | 41.3        | 35.2         |
> | FIFO                             | 43.9(+2.6)  | 38.8(+3.6)   |
> | remove closest one               | **44.1(+2.8)**  | **39.0(+3.8)**   |
>
> This improvement brings +2.8% $\rm{mAP}^{box}$ and +3.8% $\rm{mAP}^{mask}$ under the noun-pronoun distillation setting, achieving an updated SOTA performance.
>
> We will present the results in the new version of the paper.
>
> ### Weakness.2
> > Some ablation studies regarding why some loss terms are useful are missing.
>
> We thank R#u6fT for this suggestion. We provided ablation studies in which we remove $\mathcal{L}\_{\rm{align}}$ or $\mathcal{L}\_{\rm{token}}$ or both. The quantitative results are demonstrated below:
>
> Table 2: Ablations for the soft-token prediction loss and the contrastive alignment loss.
>
> |                 Method                 | $\rm{mAP}^{box}$ | $\rm{mAP}^{mask}$ |
> |:--------------------------------------:|:-----------:|:------------:|
> | TOIST                                  | **41.3**        | **35.2**         |
> | TOIST w/o $\mathcal{L}\_{\rm{token}}$                  | 40.1(-1.2)  | 34.8(-0.4)   |
> | TOIST w/o $\mathcal{L}\_{\rm{align}}$                  | *41.1(-0.2)*  | *35.1(-0.1)*   |
> | TOIST w/o $\mathcal{L}\_{\rm{token}}$ and $\mathcal{L}\_{\rm{align}}$  | 23.4(-17.9) | 20.7(-14.5)  |
>
> It shows that removing $\mathcal{L}\_{\rm{token}}$ brings a performance drop of -1.2% $\rm{mAP}^{box}$ and -0.4 $\rm{mAP}^{mask}$, because the association between the matched object predictions and the task descriptions is weakened.
> Removing $\mathcal{L}\_{\rm{align}}$ brings a performance drop of -0.2% $\rm{mAP}^{box}$ and -0.1% $\rm{mAP}^{mask}$, because the features of an object and its corresponding text features cannot be explicitly constrained to be closer.
> Interestingly, removing both of them brings a significant performance drop of -17.9% $\rm{mAP}^{box}$ and -14.5% $\rm{mAP}^{mask}$, implying the two loss terms enhance the effect of each other to make TOIST understand verb reference better.
>
> We will add the results into the supplementary.
>
> ### Question.1
> > How many unique pronouns are used in the captions to train the student model? How many objects in Coco Task dataset? Is the coco task dataset captions modified in any way (like replacing objects with pronouns) to train the student model?
>
> Before we answer questions, we use an example to illustrate how we generate text inputs:
>
> Let us consider a task whose caption is 'dig hole'.
> We use 'dig hole with pronoun' as the text input of our plain TOIST or the student TOIST, where the 'pronoun' (like 'something') is the same for all the data. We use 'dig hole with noun' for the teacher TOIST, where the 'noun' is the name of the ground truth object (like 'skateboard') that changes with the input image.
>
>
> Then we answer the questions:
>
> Firstly, we use only one unique pronoun for all verb phrases to train the student model. This unique pronoun can be 'something', 'it', 'them' or 'abcd', as demonstrated in Table.4.
>
> Secondly, the COCO-Tasks dataset contains a total of 65797 objects spanning 49 categories.
>
> Thirdly, the COCO-Tasks dataset provides captions for verb phrases separately, and we concatenate the phrase with the selected pronoun to train the student model. So there is no need to 'replacing objects with pronouns'.

---

> ### Author Response · Authors · 2022-08-02
> **Responses to Reviewer #u6fT [2/3]**
>
>
> ### Question.2
> > It would be good to see an analysis of what verbs are associated with what objects (comparing ground truth and model predictions). Something like a distribution plot that the verb "sit on" was associated with "chair" in 10 out of 20 times, it was associated with "table" in 5 out of 20 times etc. That will indicate if the model fails for any verbs more frequently than others.
>
> We thank R#u6fT for this suggestion. We have included some statistics in the Figure 1-8 in the supplementary for analysis (we mark the corresponding analysis in section 4 of the supplementary in red). This analysis sheds more light on noun-pronoun distillation. As demonstrated, we reach the conclusion that the proposed distillation method makes TOIST more capable of filtering out objects that do not afford the tasks. And the effect of the distillation on different categories is influenced by the proportion of categories in the tasks. When a few classes take a large portion of selected objects in a certain task, the effect of the distillation on these classes is good, while that on others is poor. If the number of categories in the whole task is distributed more evenly, the distillation can boost performance for most categories.
>
>
> ### Question.3
> > How is the score ${\hat s}\_{i}$ is used in loss function in equation 3? Is it used in localization loss terms or segmentation loss terms? Or is it used in some other way?
>
> ${\hat s}\_{i}$ is used in loss terms in an indirect way through the predicted logits ${\hat{\mathbf{g}}\_{i}}$.
>
> The preference score ${\hat s}\_{i}$ is defined as
> ${\hat s}\_{i} = 1 - \frac{\exp \left(\hat g\_{n\_{\rm{max}}}^i\right)}{\sum\_{j=1}^{n\_{\rm{max}}} \exp \left(\hat g\_{j}^i\right)}$, in which the predicted logits ${\hat{\mathbf{g}}\_{i}} = [\hat g\_1^i, \ldots, \hat g\_{n\_{\rm{max}}}^i]$ is constrained by the soft-token prediction loss $\mathcal{L}\_{\rm{token}}$. And $\mathcal{L}\_{\rm{token}}$ is defined as:
> $\mathcal{L}\_{\rm{token}}({{\mathbf{p}}^{\rm{span}}\_{i}}, {\hat{\mathbf{g}}\_{\sigma\_0(i)}}) = -\sum\_{j}^{n\_{\rm{max}}} p^{\rm{span}}\_{i,j} \log\frac{\exp \left(\hat g\_{j}^{\sigma\_0(i)}\right)}{\sum\_{l=1}^{n\_{\rm{max}}} \exp \left(\hat g\_{l}^{\sigma\_0(i)}\right)}$. More details about loss functions can be found in section 1.2 of the supplementary.
>
> ### Question.4
> > How is default value of $n\_{\rm{max}}$ = 256 decided? What is the value of $n\_{\rm{pred}}$? I am assuming it should be greater than the total number of objects in CoCo dataset. Is that the case?
>
> We follow MDETR [1] to set the default value of $n\_{\rm{max}}$ to be 256.
> The value of $n\_{\rm{pred}}$ is 100, which is greater than the maximum number of objects in all images of the COCO-Tasks dataset.
>
>
> ### Question.5
> > Ablations for including vs not including the loss terms $\mathcal{L}\_{\rm{token}}$ and $\mathcal{L}\_{\rm{align}}$?
>
> Please see the response to weakness.2.
>
> ### Question.6
> > It is not clear what $\mathfrak{S}\_{{n\_{\rm{pred}}}}$ means in line 245. Please define what it stands for.
>
> $\mathfrak{S}\_{{n\_{\rm{pred}}}}$ is the set of all permutations of $n\_{\rm{pred}}$ elements.
>
> For instance, all permutations of the set $S = \{1,2,3\}$ can be written as:
> $\sigma\_1=\left(\begin{array}{lll} 1 & 2 & 3 \\\\ 1 & 2 & 3 \end{array}\right)$, $\sigma\_2=\left(\begin{array}{lll} 1 & 2 & 3 \\\\ 1 & 3 & 2 \end{array}\right)$, $\sigma\_3=\left(\begin{array}{lll} 1 & 2 & 3 \\\\ 2 & 1 & 3 \end{array}\right)$, $\sigma\_4=\left(\begin{array}{lll} 1 & 2 & 3 \\\\ 2 & 3 & 1 \end{array}\right)$, $\sigma\_5=\left(\begin{array}{lll} 1 & 2 & 3 \\\\ 3 & 1 & 2 \end{array}\right)$, $\sigma\_6=\left(\begin{array}{lll} 1 & 2 & 3 \\\\ 3 & 2 & 1 \end{array}\right)$.
> Here, $\sigma\_4$ satisfies $\sigma\_4(1)=2$, $\sigma\_4(2)=3$ and $\sigma\_4(3)=1$. The same goes for others. And then $\mathfrak{S}\_{3} = \\{\sigma\_1, \sigma\_2, \dots,\sigma\_6\\}$.

---

> ### Author Response · Authors · 2022-08-02
> **Responses to Reviewer #u6fT [3/3]**
>
>
> ### Question.7
> > $\mathcal{L}\_{\rm{match}}$ was a loss term in DETR model to encourage matching the class and bounding boxes of ground truth and prediction. However it is not included in loss functions here (equations 3 and 9). Why is $\mathcal{L}\_{\rm{match}}$ not used in loss function? In line 247, authors mention KL divergence is also a part of $\mathcal{L}\_{\rm{match}}$. However, the original DETR paper doesnt mention that.
>
> Firstly, in DETR, $\mathcal{L}\_{\rm{match}}$ is used to find an optimal bipartite matching between predicted and ground truth objects. It is not a loss term for backpropagation.
>
> Secondly, in our method, we calculate the bipartite matching $\hat \sigma\_0$ with:
>
>
>
> $\hat{\sigma\_0}=\mathop{\arg \min }\limits\_{\sigma\_0 \in \mathfrak{S}\_{{n\_{\rm{pred}}}}} \sum\_{i}^{{n\_{\rm{pred}}}}
> \mathbb{1}\_{\{p^{\rm{span}}\_{i,n\_{\rm{max}}} = 0\}}
> \left[\mathcal{L}\_{\rm{l1}}(b\_{i}, \hat{b}\_{\sigma\_0(i)})+
> \mathcal{L}\_{\rm{giou}}(b\_{i}, \hat{b}\_{\sigma\_0(i)})+
> \mathcal{L}\_{\rm{token-m}}({{\mathbf{p}}^{\rm{span}}\_{i}}, {\hat{\mathbf{g}}\_{\sigma\_0(i)}})\right].$
>
>
>
> Here,
>
>
>
> $\mathcal{L}\_{\rm{l1}}(b\_{i}, \hat{b}\_{\sigma\_0(i)}) = \left\|b\_{i}-\hat{b}\_{\sigma\_0(i)}\right\|\_{1},$
>
> $\mathcal{L}\_{\text {giou}}(b\_{i}, \hat{b}\_{\sigma\_0(i)})=1-\left(\frac{|b\_{i} \cap \hat{b}\_{\sigma\_0(i)}|}{|b\_{i} \cup \hat{b}\_{\sigma\_0(i)}|}-\frac{|B(b\_{i}, \hat{b}\_{\sigma\_0(i)}) \backslash b\_{i} \cup \hat{b}\_{\sigma\_0(i)}|}{|B(b\_{i}, \hat{b}\_{\sigma\_0(i)})|}\right),$
>
> $\mathcal{L}\_{\rm{token-m}}({{\mathbf{p}}^{\rm{span}}\_{i}}, {\hat{\mathbf{g}}\_{\sigma\_0(i)}}) = -\sum\_{j}^{n\_{\rm{max}}} p^{\rm{span}}\_{i,j} \frac{\exp \left(\hat g\_{j}^{\sigma\_0(i)}\right)}{\sum\_{l=1}^{n\_{\rm{max}}} \exp \left(\hat g\_{l}^{\sigma\_0(i)}\right)}.$
>
>
>
> More details can be found in section 1.2 of the supplementary.
>
> Thirdly, $\mathcal{L}\_{\rm{match}}$ mentioned in line 245-247 is proposed to find a bipartite matching between $n\_{\rm{pred}}$ object predictions of the teacher model and $n\_{\rm{pred}}$ object predictions of the student model, which is not the same as $\mathcal{L}\_{\rm{match}}$ in DETR. And in our method, we leverage KL-Divergence for preference distillation.
>
>
>
> ### Question.8
> > Instead of minimizing the distance between $l\_{\rm{pron}}^{\rm{tr}}$ and $l\_{c\_s}^j$ in equation 4, why can't one minimize the distance between $l\_{\rm{pron}}^{\rm{tr}}$ and $l\_{\rm{noun}}^{\rm{tr}}$ for knowledge distillation?
>
> We have tried this simplified method but it does not work well. The quantitative results are demonstrated below:
>
>
> Table 3: Comparison of different distillation methods.
>
> |              Method              | $\rm{mAP}^{box}$ | $\rm{mAP}^{mask}$ |
> |:--------------------------------:|:-----------:|:------------:|
> | TOIST                            | 41.3        | 35.2         |
> | distill from $l\_{c\_s}^j$ to $l\_{\rm{pron}}^{\rm{tr}}$            | **43.9(+2.6)**  | **38.8(+3.6)**   |
> | distill from $l\_{\rm{noun}}^{\rm{tr}}$ to $l\_{\rm{pron}}^{\rm{tr}}$ | *41.9(+0.6)*  | *36.0(+0.8)*   |
>
>
>
> We will present the result in the new version of the paper.
>
>
> ### References
> [1] Kamath A, Singh M, LeCun Y, et al. MDETR-modulated detection for end-to-end multi-modal understanding[C]//Proceedings of the IEEE/CVF International Conference on Computer Vision. 2021: 1780-1790.

---

> ### Author Response · Authors · 2022-08-08
> **Response to Reviewer #u6fT**
>
> Dear reviewer,
>
> Please let us know if our responses have addressed the issues raised in your review. We hope that our corrections, clarifications, and additional results address the concerns you've raised. We are happy to address any further concerns.

---

### Meta-Review · Area_Chair_bEYx · 2022-08-26

**Recommendation:** Accept
**Confidence:** Certain

**Metareview:**

The paper proposes a Task-Oriented Instance Segmentation Transformer (TOIST) approach for finding objects that best afford a verb-indicated action, to handle the affordance recognition task. TOIST proposes two approaches of teacher-student knowledge distillation — it leverages the referring expression comprehension algorithm as the teacher module for guiding the student module to learn the noun-pronoun transformation. Experiments on Coco tasks show the gains of the proposed approach.

All the reviewers accepted the paper, however there were multiple suggestions that would be good for the authors to address. Reviewer aXr4 recommended simplifying aspects of the approach further, and recommended using a more recent baseline. Reviewer u6fT suggested adding some more ablation experiments and asked for clarifications in the loss function formulation. Reviewer mBEm suggested using a different baseline, and had concerns about the proposed method using extra training data compared to the baselines.

Based on the feedback provided by the reviewers, we recommend this paper for publication at NeurIPS 2022. We thank the authors for addressing some of the comments of the reviewers in their original review and subsequent author feedback period. The authors seem to have reported new results and addressed the concerns/feedback from the reviewers in the rebuttal period -- it would be good to include these additional results and discussions as much as possible in the updated paper/supplemental materials.

**Award:**

No

---

### Decision · Program_Chairs · 2022-09-14

Accept